# On the Geometry of Reinforcement Learning in Continuous State and Action Spaces

## Abstract

Advances in reinforcement learning have led to its successful application in complex tasks with continuous state and action spaces. Despite these advances in practice, most theoretical work pertains to finite state and action spaces. We propose building a theoretical understanding of continuous state and action spaces by employing a geometric lens. Central to our work is the idea that the transition dynamics induce a low dimensional manifold of reachable states embedded in the high-dimensional nominal state space. We prove that, under certain conditions, the dimensionality of this manifold is at most the dimensionality of the action space plus one. This is the first result of its kind, linking the geometry of the state space to the dimensionality of the action space. We empirically corroborate this upper bound for four MuJoCo environments. We further demonstrate the applicability of our result by learning a policy in this low dimensional representation. To do so we introduce an algorithm that learns a mapping to a low dimensional representation, as a narrow hidden layer of a deep neural network, in tandem with the policy using DDPG. Our experiments show that a policy learnt this way perform on par or better for four MuJoCo control suite tasks.

## 1 Introduction

The goal of a reinforcement learning (RL) agent is to learn an optimal policy that maximises the return which is the time discounted cumulative reward (Sutton & Barto, 1998). Recent advances in RL research have lead to agents successfully learning in environments with enormous state spaces, such as games (Mnih et al., 2015; Silver et al., 2016), and robotic control in simulation (Lillicrap et al., 2016; Schulman et al., 2015; 2017a) and real environments (Levine et al., 2016; Zhu et al., 2020; Deisenroth & Rasmussen, 2011). However, we do not have an understanding of the intrinsic complexity of these seemingly large problems. For example, in most popular deep RL algorithms for continuous control, the agent's policy is parameterised by a a deep neural network (DNN) (Lillicrap et al., 2016; Schulman et al., 2015; 2017a) but we do not have theoretical models to guide the design of DNN architecture required to efficiently learn an optimal policy for various environments. There have been approaches to measure the difficulty of an RL environment from a sample complexity perspective (Antos et al., 2007; Munos & Szepesvari, 2008; Bastani, 2020) but these models fall short of providing recommendations for the policy and value function complexity required to learn an optimal policy.

We view the complexity of RL environments through a geometric lens. We build on the intuition behind the *manifold hypothesis*, which states that most high-dimensional real-world datasets actually lie on low-dimensional manifolds (Tenenbaum, 1997; Carlsson et al., 2007; Fefferman et al., 2013; Bronstein et al., 2021); for example, the set of natural images are a very small, smoothly-varying subset of all possible value assignments for the pixels. A promising geometric approach is to model the data as a low-dimensional structure—a *manifold*—embedded in a high-dimensional ambient space. In supervised learning, especially deep learning theory, researchers have shown that the approximation error depends strongly on the dimensionality of the manifold (Shaham et al., 2015; Pai et al., 2019; Chen et al., 2019; Cloninger & Klock, 2020), thereby connecting the complexity of the underlying structure of the dataset to the complexity of the DNN.

As in supervised learning, researchers have applied the manifold hypothesis in RL—i.e. hypothesized that the effective state space lies on a low dimensional manifold (Mahadevan, 2005; Machado et al.,

2017; 2018; Banijamali et al., 2018; Wu et al., 2019; Liu et al., 2021). Despite its fruitful applications, this assumption—of a low-dimensional underlying structure—has never been theoretically and empirically validated in any RL setting.

Our main result provides a general proof of this hypothesis for *all* continuous state and action RL environments by proving that the effective state space is a manifold and upper bound its dimensionality by, simply, the dimensionality of the action space plus one. Although our theoretical results are for deterministic environments with continuous states and actions, we empirically corroborate this upper bound on four MuJoCo environments (Todorov et al., 2012), with sensor inputs, by applying the dimensionality estimation algorithm by Facco et al. (2017). Our empirical results suggest that in many instances the bound on the dimensionality of the effective state manifold is tight. To show the applicability and relevance of our theoretical result we empirically demonstrate that a policy can be learned using this low-dimensional representation that performs as well as or better than a policy learnt using the higher dimensional representation. We present an algorithm that does two things simultaneously: **1)** learns a mapping to a low dimensional representation, called the *co-ordinate chart*, parameterised by a DNN, and **2)** uses this low-dimensional mapping to learn the policy. Our algorithm extends DDPG (Lillicrap et al., 2016) and uses it as a baseline with a higher-dimensional representation as the input. We empirically show a surprising new DNN architecture with a bottleneck hidden layer of width equal to dimensionality of action space plus one performs on par or better than the wide architecture used by Lillicrap et al. (2016). These results demonstrate that our theoretical results, which speaks to the underlying geometry of the problem, can be applied to learn a low dimensional or compressed representation for learning in a data efficient manner. Moreover, we connect DNN architectures to effectively learning a policy based on the underlying geometry of the environment.

## 2 Background and Mathematical Preliminaries

We first describe the continuous time RL model and Markov decision process (MDP). This forms the foundation upon which our theoretical result is based. Then we provide mathematical background on various ideas from the theory of manifolds that we employ in our proofs and empirical results.

### 2.1 Continuous-Time Reinforcement Learning

We analyze the setting of continuous-time reinforcement learning in a deterministic *Markov decision process* (MDP) which is defined by the tuple $\mathcal{M} = (\mathcal{S}, \mathcal{A}, f, f_r, s_0, \lambda)$ over time $t \in [0, T]$. $\mathcal{S} \subset \mathbb{R}^{d_s}$ is the set of all possible states of the environment. $\mathcal{A} \subset \mathbb{R}^{d_a}$ is the rectangular set of actions available to the agent. $f : \mathcal{S} \times \mathcal{A} \times \mathbb{R}^+ \to \mathcal{S}$ and $f \in C^\infty$ is a *smooth* function that determines the state transitions: $s' = f(s, a, \tau)$ is the state the agent transitions to when it takes the action $a$ at state $s$ for the time period $\tau$. Note that $f(s, a, 0) = s$, meaning that the agent's state remains unchanged if an action is applied for a duration of $\tau = 0$. The reward obtained for reaching state $s$ is $f_r(s)$, determined by the reward function $f_r : \mathcal{S} \to \mathbb{R}$. $s_t$ denotes the state the agent is at time $t$ and $a_t$ is the action it takes at time $t$. $s_0$ is the fixed initial state of the agent at $t = 0$, and the MDP terminates at $t = T$. The agent does not have access to the functions $f$ and $f_r$, and can only observe states and rewards at a given time $t \in [0, T]$.

The agent is equipped with a policy, $\pi : \mathcal{S} \to \mathcal{A}$, that determines its decision making process. We denote the set of all the possible policies by $\Pi$. Simply put, the agent takes action $\pi(s)$ at state $s$. The goal of the agent is to maximise the discounted return $J(\pi) = \int_0^T e^{-\frac{l}{\lambda}} f_r(s_l) dl$, where $s_{t+\epsilon} = f(s_t, \pi(s_t), \epsilon)$ for infinitesimally small $\epsilon$ and all $t \in [0, T]$. We define the *action tangent mapping*, $g : \mathcal{S} \times \mathcal{A} \to \mathbb{R}^{d_s}$, for an MDP as

$$g(s, a) = \lim_{\epsilon \to 0^+} \frac{f(s, a, \epsilon) - s}{\epsilon} = \frac{\partial f(s, a, \epsilon)}{\partial \epsilon}.$$

Intuitively, this captures the direction in which the agents state changes at state $s$ upon taking an action $a$. For notational convenience we will denote $g(s, \pi(s))$ as $g_\pi : \mathcal{S} \to \mathbb{R}^{d_s}$ and name it the *action flow* of the environment defined for a policy $\pi$. Note that $g_\pi$ is a well defined function. Intuitively, $g_\pi$ is the direction of change in the agent's state upon following a policy $\pi$ at state $s$ for an infinitesimally small time. The curve in the set of possible states, or the state-trajectory of the agent, is a differential

equation whose integral form is as follows,

$$s_t^\pi = s_0 + \int_0^t g_\pi(s_l^\pi)dl. \tag{1}$$

This solution is also unique (Wiggins, 1989) for a fixed start state, $s_0$, and policy, $\pi$. The above curve is a smooth curve if the policy is also smooth. Therefore, given an MDP, $\mathcal{M}$, and a smooth deterministic policy, $\pi \in \Pi$, the agent traverses a continuous time state-trajectory or curve $H_{\mathcal{M},\pi} : [0, T) \to \mathcal{S}$.

The value function at time $t$ for a policy $\pi$ is the cumulative future reward starting at time $t$:

$$v^\pi(s_t) = \int_t^T e^{-\frac{l-t}{\lambda}} f_r(s_l^\pi)dl. \tag{2}$$

Note that the objective function, $J(\pi)$, is the same as $v^\pi(s_0)$. Our specification is very similar to classical control and continuous time RL (Cybenko, 1989; Doya, 2000) with the key difference being how we define the transition function, $f$.

## 2.2 MANIFOLDS

MDPs, in practice, have a low-dimensional underlying structure resulting in them having fewer degrees of freedom than their nominal dimensionality. In the Cheetah MujoCo environment, with image observations, the goal of the RL agent is to learn a policy to make the Cheetah move forward as fast as possible, where the Cheetah is constrained to a plane. The actions available to the agent are providing torques at each one of the 6 joints. In the case of learning from visual input in a MuJoCo environment like 2D cheetah, one can describe the cheetah's state by its "pose" and position instead of the $128 \times 128$ pixels of the image. The idea of a low dimensional manifold embedded in a high dimensional state space formalises this.

A function $h : X \to Y$, from one open subset $X \subset \mathbb{R}^{l_1}$, to another open subset $Y \subset R^{l_2}$, is a diffeomorphism if $h$ is bijective, and both $h$ and $h^{-1}$ are differentiable. Intuitively, a low dimensional surface embedded in a high dimensional Euclidean space can be parameterised by a differentiable mapping, and if this mapping is bijective we term it a diffeomorphism. Here $X$ is said to be diffeomorphic to $Y$. A manifold is defined as follows.

**Definition 2.1.** *A subset $M \subset \mathbb{R}^k$ is called a smooth $m$-dimensional submanifold of $\mathbb{R}^k$ (or $m$-manifold in $\mathbb{R}^k$) iff every point $p \in M$ has an open neighborhood $U \subset \mathbb{R}^k$ such that $U \cap M$ is diffeomorphic to an open subset $O \subset \mathbb{R}^m$. A diffeomorphism, $\phi : U \cap M \to O$ is called a coordinate chart of $M$ and the inverse, $\psi := \phi^{-1} : O \to U \cap M$ is called a smooth parameterisation of $U \cap M$.*

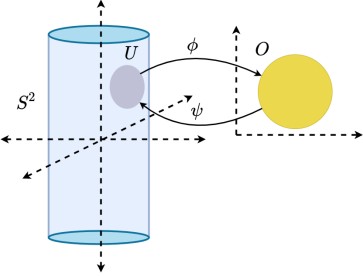

Figure 1: The surface of an open cylinder of unit radius, denoted by $S^2$, in $\mathbb{R}^3$ is a 2D manifold embedded in a 3D space. More formally, $S^2 = \{(x, y, z)|x^2 + y^2 = 1, z \in (-h, h)\}$ where the cylinder's height is $2h$. One can smoothly parameterise $S^2$ as $\psi(\theta, b) = (\sin\theta, \cos\theta, b)$. The coordinate chart is $\phi(x, y, z) = (\sin^{-1} x, z)$.

We illustrate this with an example in Figure 1. If $M \subset \mathbb{R}^k$ is a non-empty smooth $m$-manifold then $m \leq k$, reflecting the idea that a manifold is of lower or equal dimension than its ambient space. A smooth curve $\gamma : I \to M$ is defined from an interval $I \subset \mathbb{R}$ to the manifold $M$ as a function that is infinitely differentiable for all $t$. The derivative of $\gamma$ at $t$ is denoted as $\dot{\gamma}(t)$. The length of a curve $\gamma : I \to M$ is defined as $L(\gamma) = \int_I ||\dot{\gamma}(t)||dt$, where $|| \cdot ||$ denotes the vector norm. In the Euclidean

space the distance between two points is the length of the unique straight line connecting them. Similarly, the *geodesic distance* between two points $a, b \in M$ is defined as the length of the shortest such curve, $\gamma : [0, 1] \to M$, such that $\gamma(0) = a$ and $\gamma(1) = b$. We denote the geodesic distance between $a, b \in M$ on the manifold $M$ is denoted by the function $d_M(a, b)$. The set of derivatives of the curve at time $t$, $\dot{\gamma}(t)$, for all possible smooth $\gamma$, form a set that is called the tangent space. The tangent space characterises the geometry of the manifold and it is defined as follows.

**Definition 2.2.** *Let $M$ be an $m$-manifold in $\mathbb{R}^k$ and $p \in M$ be a fixed point. A vector $v \in \mathbb{R}^k$ is called a tangent vector of $M$ at $p$ if there exists a smooth curve $\gamma : I \to M$ such that $\gamma(0) = p, \dot{\gamma}(0) = v$. The set $T_p M := \{\dot{\gamma}(0) | \gamma : \mathbb{R} \to M \text{ is smooth}, \gamma(0) = p\}$ of tangent vectors of $M$ at $p$ is called the tangent space of $M$ at $p$.*

Continuing our example, the tangent space of a point $p$ in $S^2$ is the vertical plane tangent to the cylinder at that point. For a small enough $\epsilon$ and a vector $v \in T_p S^2$ there exists a unique curve $\gamma : [-\epsilon, \epsilon] \to S^2$ such that $\gamma(0) = p$ and $\dot{\gamma}(0) = v$.

## 3 STATE SPACE GEOMETRY

The state space is typically thought of as a dense Euclidean space in which all states lie, but it is not necessarily the case that all such states are reachable by the agent. Two main factors constrain the states available to an agent: **1)** the transition function and the actions that are available to an agent, and **2)** the start state $s_0$. We therefore define $\mathcal{S}_e$ as the *effective set of states*. Under the assumptions that $\Pi$ is the set of all smooth policies, $\pi : \mathcal{S} \to \mathcal{A}$, for a continuous time MDP, $\mathcal{M}$, and those made in section 2.1 we define the effective set of states as follows.

**Definition 3.1.** *For an MDP, $\mathcal{M}$, the effective set of states, $\mathcal{S}_e$, is defined as the union of the sets of states of all possible continuous curves traversed by the agent for the set of all smooth policies. Formally, $\mathcal{S}_e = \cup_{\pi \in \Pi} \{s | s = H_{\mathcal{M}, \pi}(t) \text{ for some } t \in (0, T)\}$, where $\Pi$ is the set of everywhere smooth policies with domain $\mathcal{S}$, and $H_{\mathcal{M}, \pi} : [0, t) \to \mathcal{S}$ is the curve defined by the policy $\pi$ given the MDP $\mathcal{M}$.*

We make three additional assumptions:

1. **Full rank Jacobian:** $f$ has a full rank Jacobian, in variables $[a, t]$, for all $s \in \mathcal{S}, a \in \mathcal{A}$ and $t \in (0, T)$, and for a fixed policy $\pi$ the solution for the equation $s_t^\pi = s$, if it exists, is unique in $t$.

2. **No small orbits:** there exists an $\tau > 0$ such that for all $t < \tau$ and a fixed policy $\pi$ the solution for the equation $s_t^\pi = s$, if it exists, is unique in $t$.

3. **Action space restriction:** there exists an $r > 0$ and an open restriction $\mathcal{A}'_s$ of $\mathcal{A}$, for every state $s \in \mathcal{S}_e$, such that $\mathcal{A}'_s \subset \mathcal{A}$ and for all $0 < \epsilon < r$, if we have $f(s, a_1, \epsilon) = f(s, a_2, \epsilon)$ for some fixed $a_1, a_2 \in \mathcal{A}'$ then $a_1 = a_2$.

We provide further explanations for these assumptions and how they restrict our study in Appendix A. Under these two assumptions for the setting of continuous RL as in Section 2.1 we have the result stated below.

**Theorem 3.2.** *The set of effective states, $\mathcal{S}_e$, is a smooth manifold of dimensionality at most $d_a + 1$.*

We provide the proof in Appendix B. Intuitively, the action set limits the directions in which the agent can go. For example, a single action executed for a finite time interval makes the agent traverse a 1D curve. The union of all such curves is the only subset of the state space the agent can reach. The key technical idea is to construct a coordinate chart, and therefore a diffeomorphism, from a subset of low dimensional euclidean space to a subset of the state space. We fix a state $s$ and consider the function $\phi : \mathcal{A} \times (0, r) \to \mathcal{S}$ such that $\phi(a, \epsilon) = f(s, a, \epsilon)$. This result pertains to the *global geometry* of the state manifold, meaning the state manifold everywhere has this low dimensional structure.

Our result formalises the long-held idea that there is in fact a lower dimensional structure to the state manifold; in particular, this is true when $d_a + 1 < d_s$, which is almost always the case for RL environments. It also formalises the idea that the agent can only observe data for states reachable by interacting with the environment using the set of actions at its disposal. Henceforth, we refer to $\mathcal{S}_e$ as the state manifold. We also present an immediate corollary of Theorem 3.2.

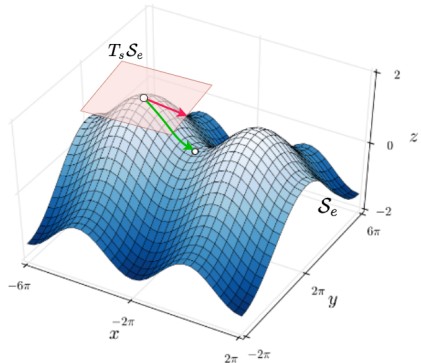

Figure 2: Consider a hypothetical scenario as in the image above where upon taking an action even though the agent moves locally at state $s$ in the direction of the red arrow (along $g_\pi(s)$) as time progresses it ends up travelling along the green arrow which is on the manifold $\mathcal{S}_e$. There is some constraint that restricts the state space "available" to an agent. Thus, the agent can only reach the states that have a valid solution for the Equation 1 for any smooth policy.

**Corollary 3.3.** *For every $a \in \mathcal{A}$ and fixed $s \in \mathcal{S}_e$, the action tangent mapping, $g(s, a)$, maps from the set of actions $\mathcal{A}$ to $T_s\mathcal{S}_e$ and therefore there exists a vector $v_a \in T_s\mathcal{S}_e$ such that $v_a = g(s, a)$ for all $a \in \mathcal{A}$.*

This implies that locally an agent is moving along a tangent in the tangent space of a state in the state manifold as it acts upon the environment. The corollary follows from Definition 2.2, and its proof is in Appendix B. Intuitively, Corollary 3.3 implies that locally the agent's state changes in a constrained manner dictated by the actions available to it and the *local geometry* of the manifold, i.e., the tangent space and its projection onto the state manifold. We illustrate this in Figure 2. We note that the idea of actions mapping to the tangent space was assumed earlier by Liu et al. (2021) for constrained RL but Corollary 3.3 both proves this assumption and shows that it holds in general for continuous RL environments.

## 4 Connections Between Continuous Time Deterministic RL and Empirical RL

We have presented our results in the continuous time RL setting, which is an underutilized theoretical tool in the study of RL. Here, and in the experimental section, we argue that it is in fact a useful model for theoretical analysis in the continuous state and action, *discrete* time, setting. The primary intuition is that, in the context of simulated robotic control problems, our work approximates the agents behavior during the transition period $(t, t + 1)$ as the action $a_t$ changes to $a_{t+1}$ in a smooth manner and similarly for $s_t$ to $s_{t+1}$. In this context, the discrete-time observations can be viewed as time-uniform samples from an underlying continuous time process. This is in keeping with with robotic control, in a robot with various joints we can only measure the joints over intervals which are then attributed to discrete time measurements. Similarly, we can actuate the motors up to a frequency and those can be considered as discrete control signals, even though the underlying physical process is continuous time.

More formally, consider a discrete time MDP with continuous state and action spaces, a discrete state trajectory is the sequence of states $\{s_t\}_{t=1}^T$ and similarly for actions, $\{a_t\}_{t=1}^T$. Let $\vartheta : \mathcal{A} \times [0, 1] \times \mathcal{A} \to \mathcal{A}$ be a function that acts as a smooth transformation between two successive actions $a_t, a_{t+1}$ such that the agent transition from $s_t$ to $s_{t+1}$ in the continuous time model, as described in Section 2.1. In other words, $\vartheta(a_t, l, a_{t+1}) = a_{t+l}$ is the discrete to continuous time transformation of actions. We postulate that there exists an operator $H_\vartheta$, dependent on the MDP $\mathcal{M}$, such that a discrete trajectory can be transformed into a continuous time trajectory. Proving such an augmentation exists, given that the underlying physical process as described in Section 2.1 and the discrete trajectory is sampled at discrete time intervals as described above, is beyond the scope of our current work. Intuitively, since the discrete time trajectories are temporally spaced out measurements of continuous trajectories the existence of such an operator can be considered an "inversion" of the sampling process.

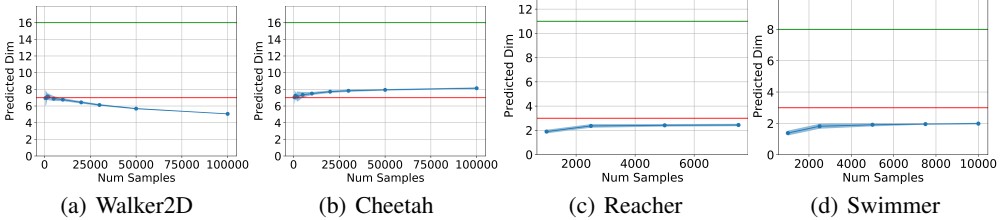

(a) Walker2D  (b) Cheetah  (c) Reacher  (d) Swimmer

Figure 3: State manifold dimensionality, in blue, is close to $d_a + 1$ (horizontal red line) and far below $d_s$ (horizontal green line) for various environments, estimated using the algorithm by Facco et al. (2017).

Finally, we address the assumption of deterministic transitions. For our empirical analysis we use the popular MuJoCo environments for robotic control with sensor inputs (Todorov et al., 2012) and these environments are deterministic for all practical intents and purposes (see Appendix C). We argue further that our theoretical model has broader applicability. One way to model transitions is to assume that the underlying state transitions are deterministic but the observations the agent receives have additive noise. More formally, a simplistic way to model stochastic transition $F(s, a, \epsilon) = f(s, a, \epsilon) + \mathcal{N}(0, \sigma)$ is the stochastic transition function such that there is additive noise to this deterministic transition. We postulate that the effective set of states can be modeled as a manifold with each observation the agent makes lying at a certain distance to the state manifold and the distance is normally distributed. This is also the idea behind many manifold learning paradigms: the data lies at a distance to a low-dimensional manifold and this distance is distributed according to some probability law (Fefferman et al., 2013; Pai et al., 2019).

## 5 EMPIRICAL VALIDATION

Our empirical validation is two fold. First, we show that the bound on the manifold dimensionality as in Theorem 3.2 holds in practice. Second, we demonstrate the practical relevance of our result by learning a "bottleneck" representation using which an RL agent can learn effectively.

### 5.1 EMPIRICAL DIMENSIONALITY ESTIMATION

To empirically corroborate our main result (Theorem 3.2) we perform experiments in the MuJoCo domains provided in the OpenAI Gym (Brockman et al., 2016). These are all continuous state and action spaces with $d_a < d_s$ for simulated robotic control tasks. The states are typically sensor measurements such as angles, velocities or orientation, and the actions are torques provided at various joints. We estimate the dimensionality of the state manifold $\mathcal{S}_e$. To sample data from the manifold, we record the trajectories from multiple evaluation runs of DDPG across different seeds (Lillicrap et al., 2016), with two changes: we use GELU activation (Hendrycks & Gimpel, 2016) instead of ReLU, in both policy and value networks, and also use a single hidden layer network of width 400 instead of 2 hidden layers for both the networks. This is inline with the assumptions for Theorem 3.2 in Section 3, that the policy is smooth. Performance is comparable to the original DDPG setting (see the Appendix G). For background on DDPG refer to Appendix F. We then randomly sample states from the evaluation trajectories to obtain a subsample of states, $\mathcal{D} = \{s_i\}_{i=1}^n \subset \mathcal{S}_e$. We estimate the dimensionality with 10 different subsamples of the same size to provide an error region for the estimates.

We employ the dimensionality estimation algorithm introduced by Facco et al. (2017), which estimates the intrinsic dimension of datasets characterized by non-uniform density and curvature, to empirically corroborate Theorem 3.2. Further details about the dimensionality estimation procedure are presented in Appendix E. The estimates for four MuJoCo environments are shown in Figure 3. For all environments the estimate remains in the neighbourhood of $d_a + 1$ in keeping with Theorem 3.2.

### 5.2 LEARNING VIA THE LOW-DIMENSIONAL MANIFOLD REPRESENTATION

We now validate the relevance of Theorem 3.2 using a popular policy gradient method, deep deterministic policy gradient (DDPG) Lillicrap et al. (2016), which is discrete time. DDPG is a framework

for learning in environments with continuous action spaces, and the policy and value function are parameterised by DNNs. Let $\theta^\pi$ be the parameters of the policy DNN. The agent learns by updating the policy parameters with respect to the discrete time discounted return, $J(\theta^\pi)$:

$$\theta^\pi \leftarrow \theta^\pi + \alpha_\pi \nabla_{\theta^\pi} J(\theta^\pi),$$

where $\alpha_\pi$ is the learning rate for the policy and $J(\theta^\pi)$ is the discounted return objective. For further details and background on DDPG please see Appendix F. We do so by learning mapping to a low dimensional manifold of size $d_a + 1$ from the control input space that feeds into the policy. The central idea is to show that this compressed representation can be used to learn a control policy with RL without any loss—and possibly even a gain— of performance.

Our goal in validating the applicability of this low dimensional representation is to learn an *isometric* coordinate chart from the high dimensional representation for states in $\mathcal{S}_e$ to a low dimensional Euclidean space $\mathbb{R}^m$, where $m \leq d_a + 1$ as noted in Theorem 3.2, and then compare learning a policy using this compression of the state to learning from the full state. We denote the coordinate chart by $\psi : \mathcal{S}_e \to \mathbb{R}^{d_a+1}$ parameterised by $\theta^\psi$. For a coordinate chart to be isometric it must preserve the geodesic distance on the manifold $\mathcal{S}_e$, meaning for $s_1, s_2 \in \mathcal{S}_e$ we have that $d_{\mathcal{S}_e}(s_1, s_2) = ||\psi(s_1; \theta^\psi) - \psi(s_2 \theta^\psi)||_2$, where $d_{\mathcal{S}_e}(s_1, s_2)$ is the geodesic distance between $s_1$ and $s_2$ on the manifold $\mathcal{S}_e$ and $|| \cdot ||_2$ denotes the Euclidean norm. Note that imposing isometry on a coordinate chart is a stronger condition than it being a diffeomorphism because it is distance preserving, in addition to being a bijection and differentiable. Such isometric mappings have been used in the past to learn the underlying manifold for high-dimensional data (Tenenbaum et al., 2000; Basri & Jacobs, 2017; Pai et al., 2019). This is done to ensure tractability of the objective that we introduce below. Given a dataset of states $\mathcal{D} = \{s_i\}_{i=1}^N \subset \mathcal{S}_e$ we minimize the loss

$$L_\psi(\theta^\psi) = \frac{1}{N} \sum_{s_i, s_j \in \mathcal{D}} \left( d_{\mathcal{S}_e}(s_1, s_2) - ||\psi(s_1; \theta^\psi) - \psi(s_2 \theta^\psi)||_2 \right)^2, \tag{3}$$

which is similar to the loss for learning low-dimensional manifold embeddings introduced by Tenenbaum et al. (2000). The computation and estimation of the geodesic distance, $d_{\mathcal{S}_e}$, is particularly challenging. One approach is to use a graph, as a discrete approximation of a manifold, to calculate this geodesic distance (Yan et al., 2007; Dong et al., 2011). The graph is constructed with each datapoint as a node and an edge in between two datapoint if one of them is $k_{nn}$-nearest neighbor of the other in the dataset. In addition to that there is a distance attribute associated with every edge that is the Euclidean distance between the two points. Therefore, the geodesic distance between two datapoints can be approximated the sum of these edge-wise attributes for the edges constituting the shortest path between the two points. Another practical challenge in the empirical estimation of the loss $L_\psi$ is sampling pairs of states for which we can obtain approximate geodesic distances with reasonable compute time. In summary, we calculate $d_{\mathcal{S}_e}$ in three steps: **1)** sample data from replay buffer and therefore the manifold $\mathcal{S}_e$, **2)** construct a $k_{nn}$-nearest neighbor graph with edge attributes, and **3)** estimate the distance between two states using the sum of edge attributes of the shortest path between these edges. The exact procedure employed by us is detailed in Appendix H and illustrated with a simple example in Figure 10, further algorithmic details are also provided in Appendix I.

Although there has been significant work in learning isometric mappings (Pai et al., 2019; Basri & Jacobs, 2017) and embeddings (Tenenbaum et al., 2000; Zha & Zhang, 2003), for data sampled from a manifold, these prior works assume that the data distribution remains static throughout the training process. However, the state distribution changes with the policy in RL and this mapping feeds into the policy itself, effecting the agents performance. We learn this isometric mapping, $\psi$, in tandem with the policy to account for the distribution shift. To do so we introduce an intermediate hidden layer of size $d_a + 1$ in the policy DNN. The output of this layer is then trained by performing gradient descent on the loss $L_\psi$. In this architecture $\theta^\psi \subset \theta^\pi$ and the gradient is as follows:

$$\theta^\pi \leftarrow \theta^\pi + \alpha_\pi \nabla_{\theta^\pi} J(\theta^\pi) - \alpha_\psi \nabla_{\theta^\pi} L_\psi \theta^\psi, \tag{4}$$

where $\alpha_\pi$ and $\alpha_\psi$ are the learning rates for the policy and the coordinate chart respectively. Note that update for parameters $\theta^\psi$ is then $\theta^\psi \leftarrow \theta^\psi - \alpha_\psi \nabla_{\theta^\psi} L_\psi \theta^\psi$, in addition to the update with respect to the objective for increasing the discounted return (See Appendix I and Algorithm 1 for further details). This is because for all $\theta_i \in \theta^\pi \setminus \theta^\psi$ we have $\nabla_{\theta_i} L(\theta^\psi) = 0$. We illustrate the architecture and how the representation feeds into the loss, for the policy network, in Figure 4. Note that the architecture and gradient updates for the DNN parameterising the $Q$ function remains the same as

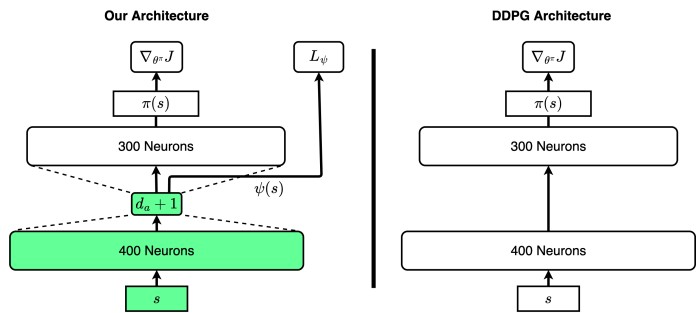

Figure 4: We introduce a bottleneck hidden layer of width $d_a + 1$ which is the output of the coordinate chart, $\psi$, i.e. the green colored base of the DNN. The output of this coordinate chart is fed into the manifold loss (Equation 3).

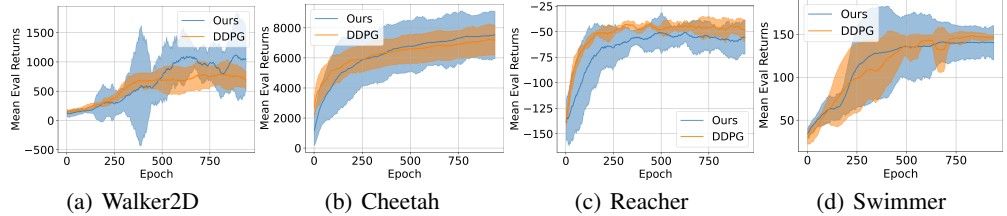

(a) Walker2D         (b) Cheetah         (c) Reacher         (d) Swimmer

Figure 5: For all the environments we use $\alpha_\psi = 10^{-5}$, in comparison to $\alpha_\pi = 10^{-4}$, and the rest of the hyper-parameters are the same as reported by Lillicrap et al. (2016), over 6 random seeds.

used by Lillicrap et al. (2016), for control inputs, which is a two hidden layer DNN with 400 neurons in the first hidden layer, 300 in the second one and a one dimensional output.

We present results for 4 different MuJoCo environments: Cheetah, Walker2D, Swimmer and Reacher in Figure 5 and for three out of four of these environments the performance is either the same or better than the DDPG baseline. Our results strongly suggest that a compressed isometric representation of dimesionality $d_a + 1$, learned using gradient updates can be used for learning a policy. This furthers our argument: there is a low dimensional structure to RL problems and it can be employed to learn more efficiently. Algorithmic details are given in Appendix I. Ablation studies for the newly introduced hyper-parameter $\alpha_\psi$ in Appendix J and comparison to training in absence of manifold loss is provided in Appendix M, due to space limitations. In addition to this, we observe that the manifold loss (Equation 3) also drops steadily as training progresses, meaning that the representation being learnt, $\psi(s; \theta_\psi)$, is a low-dimensional isometric representation which retains the manifold geometry. The graphs for how the manifold loss, $L_\psi$, evolves in each case is given in Figure 6 along with the interpretation. We explain the reasons for Reachers' failure for our algorithm in Appendix K. In summary, to learn an isometric mapping to the Euclidean space the agent might require more than $\dim(\mathcal{S}_e)$ dimensions, for the case of Reacher. The additional constraint that we place, that the learnt coordinate chart has to be isometric, is detrimental in this case. As has been done in manifold learning literature, we need methods for learning the low-dimensional mapping that do not rely on learning an isometry (Roweis & Saul, 2000; Zhang & Zha, 2004). We demonstrate the effects of changing the width of the bottleneck layer on the Cheetah domain in Figure 12, in the Appendix. We also need better methods to approximate the geodesic distance between data points in the very high-dimensional setting. We also report additional results for learning with soft actor critic algorithm (Haarnoja et al., 2018) in conjunction with manifold representation learning in Appendix L.

## 6 RELATED WORK

There has been significant empirical work that assumes the set of states to be a manifold in RL. The primary approach has been to study discrete state spaces as data lying on a graph which has an underlying manifold structure. Mahadevan & Maggioni (2007) provided the first such framework to utilise the manifold structure of the state space in order to learn value functions. Machado et al. (2017) and Jinnai et al. (2020) showed that PVFs can be used to implicitly define options and applied

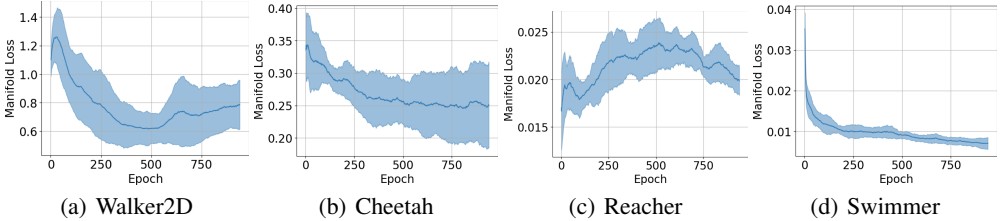

(a) Walker2D  (b) Cheetah  (c) Reacher  (d) Swimmer

Figure 6: We observe that the the loss $L_\psi$ gradually decreases in all instances except Reacher where the performance is sub-par. For the Walker2D environment we see an increase in manifold error that matches the downward spike in performance in Figure 5 but the return for our method still surpasses the baseline in this case.

them to high dimensional discrete action MDPs (Atari games). Wu et al. (2019) provided an overview of varying geometric perspectives of the state space in RL and also show how the graph Laplacian is applied to learning in RL. Another line of work, that assumes the state space is a manifold, is focused on learning manifold embeddings or mappings. Several other methods apply manifold learning to learn a compressed representation in RL (Bush & Pineau, 2009; Antonova et al., 2020; Liu et al., 2021). Jenkins & Mataric (2004) extend the popular ISOMAP framework (Tenenbaum, 1997) to spatio-temporal data and they apply this extended framework to embed human motion data which has applications in robotic control. Bowling et al. (2005) demonstrate the efficacy of manifold learning for dimensionality reduction for a robot's position vectors given additional neighbourhood information between data points sampled from robot trajectories. At the same time, continuous RL has been applied to continuous robotic control (Doya, 2000; Deisenroth & Rasmussen, 2011; Duan et al., 2016). We apply continuous state, action and time RL as a theoretical model to study the geometry of popular continuous RL environments for the first time.

More recently, in various papers that take a theoretical approach to deep learning the intrinsic dimension of the data manifold and its geometry play an important role in determining the complexity of the learning problem (Shaham et al., 2015; Cloninger & Klock, 2020; Goldt et al., 2020; Paccolat et al., 2020; Buchanan et al., 2021). Schmidt-Hieber (2019) shows that, under assumptions over the function being approximated, the statistical risk deep ReLU networks approximating a function can be bounded by an exponential function of the manifold dimension. Basri & Jacobs (2017) theoretically and empirically show that SGD can learn isometric maps from high-dimensional ambient space down to $m$-dimensional representation, for data lying on an $m$-dimensional manifold, using a two-hidden layer neural network with ReLU activation where the second layer is only of width $m$. Similarly, Ji et al. (2022) show that the sample complexity of off-policy evaluation depends strongly on the intrinsic dimensionality of the manifold and weakly on the embedding dimension. Coupled with our result, these suggest that the complexity of RL problems and data efficiency would be influenced more by the dimensionality of the state manifold, which is upper bounded by $d_a + 1$, as opposed to the ambient dimension. Finally, our work could be applied in conjunction with recent work that studies RL algorithms in light of the underlying structure of deep $Q$-functions (Kumar et al., 2021).

## 7 DISCUSSION AND CONCLUSION

We have shown that that the dimensionality of the manifold is upper bounded by $d_a + 1$ and have empirically verified it (Figure 3). This proves that the popular manifold assumption (Mahadevan, 2005; Machado et al., 2017; 2018; Banijamali et al., 2018; Wu et al., 2019; Liu et al., 2021) holds under certain conditions in continuous-time reinforcement learning. It also shows that there is an underlying lower dimensional structure to the MDPs. Additionally, we demonstrate the applicability of this result in a practical setting by showing that a DDPG agent can learn efficiently in this highly compressed, low-dimensional space. Our newly introduced architecture and simultaneous low-dimensional representation learning along with policy learning performs on par or better than the baseline DDPG approach, for MuJoCo environments with sensor inputs. Overall, we show a theoretical bound on intrinsic dimensionality of continous RL problem and also show the efficacy of this low-dimensional representation in learning a policy. This opens up room for new theoretical and empirical advances paving way for better DNN architecture design and representation learning algorithms.

## 8 REPRODUCIBILITY STATEMENT

We offer the following details for all the experiments we have performed, in the main body or appendix: hyperparameters, sample sizes, GPU-hours, CPU-hours, code, neural network architectures, Python libraries used, input sizes, and external code bases. We also provide the code with instructions on running it in the supplementary material. All the details to run the code and its location are in Appendix O. All the hyperparameters, for our experiments in Section 5.2, can be found in appendices I, J and K.

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

## A  ASSUMPTIONS

We list all the assumptions by section and consequently by theorems. We also provide some intuition on what these assumptions mean and how they limit our study.

Assumption made in Section 2.1 about continuous-time RL:

1. We assume that the transition function, $f$, is infinitely differentiable or smooth. This is done to ensure that the manifold it forms is also smooth. This might not always hold in practice but the transitions can be $C^k$ smooth, meaning $k-$times differentiable. We use this assumption in the construction of a diffeomorphism in the definition of a manifold.

2. The set of actions, $\mathcal{A}$, is rectangular and open. This is pretty standard in practice for continuous environments. This much less an assumption but done so to simplify the analysis and avoid any confusion. The only reason we need the open part is to construct a diffeomorphism.

We do not state the entirety of the assumptions made in Section 3 but we provide further intuition for them:

1. We also use assumption that the Jacobian of $f$ is full rank, in $[a, t]$. This assumption is needed for applying the implicit function theorem in the proof of Proposition B.2.

2. The no small orbits assumption is a strong one but it is essential to proving Theorem 3.2. This might not hold strictly in practice but it can be argued that it holds almost always with probability 1, essentially the probability of an agent making an orbit is almost surely 0. Moreover, as long as there is a positive $\tau$ such that there are no orbits for time $t < \tau$ this condition is satisfied. In other words, this condition is akin to saying there is no reversibility, of state transitions, in infinitesimally small time intervals. It is naturally true in environments with physical dynamics, e.g., acceleration, velocity or displacement cannot be instantaneously reversed.

3. The action space restriction condition helps us construct a bijection for the proof of Theorem 3.2. Intuitively, the idea is that if two actions have the same local effect on how the agents state changes in some state then they can be collapsed into a single action for a fixed state $s$.

## B  PROOF OF THEOREM 3.2

We first state the implicit function theorem (for more details see theorem 3.3.1 in Chapter 3 by Krantz & Parks (2002) and its proof using the inverse function theorem). In the statement of the implicit function theorem we use $F$ to denote a general function, that satisfy the conditions stated in the theorem.

**Theorem B.1.** *(**Implicit Function Theorem**) Let $F : \mathbb{R}^{d_1+d_2} \to \mathbb{R}^{d_2}$ be a continuously differentiable function. Let $\mathbb{R}^{d_1+d_2}$ have coordinates $(\boldsymbol{x}, \boldsymbol{y})$ and fix a point $(\boldsymbol{x}_0, \boldsymbol{y}_0) = (x_{0,1}, ..., x_{0,d_1}, y_{0,1}, ..., y_{0,d_2})$ with $F(\boldsymbol{x}_0, \boldsymbol{y}_0) = \boldsymbol{0}$, where $\boldsymbol{0} \in \mathbb{R}^{d_2}$. The appropriate Jacobian is defined as:*

$$J_{F, \boldsymbol{y}}(\boldsymbol{x}, \boldsymbol{y}) = \begin{bmatrix} \frac{\partial F_1}{\partial y_1}(\boldsymbol{x}_0, \boldsymbol{y}_0) & \cdots & \frac{\partial F_1}{\partial y_{d_2}}(\boldsymbol{x}_0, \boldsymbol{y}_0) \\ \vdots & \ddots & \vdots \\ \frac{\partial F_{d_2}}{\partial y_1}(\boldsymbol{x}_0, \boldsymbol{y}_0) & \cdots & \frac{\partial F_{d_2}}{\partial y_{d_2}}(\boldsymbol{x}_0, \boldsymbol{y}_0) \end{bmatrix}.$$

*If this Jacobian matrix is invertible then there exists an open set $V \subset \mathbb{R}^{d_1}$ containing $\boldsymbol{x}_0$ such that there exists a uniquely continuous differentiable function $h : V \to \mathbb{R}^{d_2}$ such that $h(\boldsymbol{x}_0) = \boldsymbol{y}_0$, and $F(\boldsymbol{x}, h(\boldsymbol{x})) = 0$ for all $\boldsymbol{x} \in V$. Additionally, $h^{-1}$ is continuously differentiable in the $h$-image of $V$.*

Using the implicit function theorem we first prove our main result for the exact case where the dimensionality of the effective state space, $\mathcal{S}_e$, is $d_a + 1$. We do so under additional assumptions in the following proposition.

**Proposition B.2.** *Under the assumptions of Section 2.1 and 3 with the added assumption that the transition function, $f$, is injective in $\mathcal{A} \times (0, L)$, for some $L$, for fixed $s \in \mathcal{S}$ the effective state space, $\mathcal{S}_e \subset \mathbb{R}^{d_s}$, is a manifold of is $d_a + 1$.*

*Proof.* Given any state $s \in \mathcal{S}_e$ we can construct an open neighbourhood that maps to an open subset of $\mathbb{R}^{d_a+1}$. Since we know that $s \in \mathcal{S}_e$ there exists a state $s' \in \mathcal{S}_e$ such that $f(s', a, \epsilon) = s$ for some $a \in \mathcal{A}$ and $\epsilon \in (0, L)$. Now, for a fixed $s'$ consider the map $\psi_{s'} : \mathcal{A} \times (\epsilon - \eta, \epsilon + \eta) \to \mathcal{S}_e$ such that $\psi_{s'}(a, t) = f(s', a, t)$ where $0 < \eta < \epsilon$, such that $\epsilon + \eta < L$, and $a \in \mathcal{A}$.

We need to show that this map, $\psi_{s'}$, is a diffeomorphism and maps to an open subset of $S_e$. Since the transition function, $f$, is smooth we have that $\psi_{s'}$ is also smooth in its range. Also note that $f$ is injective $\mathcal{A} \times (0, L)$ for fixed $s'$ and therefore it maps $\mathcal{A} \times (\epsilon - \eta, \epsilon + \eta)$ uniquely to a set $U$. This set $U$ is a subset of $\mathcal{S}_e$ by definition. Note that $U$ is open because $\mathcal{A} \times (\epsilon - \eta, \epsilon + \eta)$ is open because a bijection maps open sets to open sets. This leads to the conclusion that $\psi_{s'} : \mathcal{A} \times (\epsilon - \eta, \epsilon + \eta) \to U \subset \mathcal{S}_e$ is a smooth bijection. Thereby providing us a smooth parameterisation from $\mathbb{R}^{d_a+1}$ to $U$. This means that its inverse, $\psi_{s'}^{-1}$ exists. Now to show that this inverse is differentiable we apply the implicit function theorem.

Let $F(\mathbf{x}, \mathbf{y}, t) = \mathbf{x} - \psi_{s'}(\mathbf{y}, t)$, note that $s'$ is fixed, and as in above the variables are $\mathbf{x}, \mathbf{y}$ and $t$. The function $F$ is restricted to the domain such that $x \in \mathcal{S}_e$, $y \in \mathcal{A}$ and $t \in (\epsilon - \eta, \epsilon + \eta)$. Since the Jacobian of $f$ is full rank, by assumption, the appropriate Jacobian, $J_{F,[\mathbf{y},t]}$ as defined in Theorem B.1, is invertible and $F$ is also continuously differentiable. Similarly, the condition $F(\mathbf{x}, \mathbf{y}, t) = \mathbf{0}$ holds for $\mathbf{x} = s$, $\mathbf{y} = a$ and $t = \epsilon$. Therefore, by the implicit function theorem there exists a unique function $h$ and an open neighbourhood $V$ containing $[a, \epsilon] \in \mathbb{R}^{d_a+1}$ such that $F(h(\mathbf{y}, t), \mathbf{y}, t) = 0$ and $h$ is differentiable. Since $h$ is unique we need to show that $h = \psi_{s'}$ for the domain $V \cap (\mathcal{A} \times (\epsilon - \eta, \epsilon + \eta))$.

The proof follows from uniqueness of $h$. We have, for the domain $V \cap (\mathcal{A} \times (\epsilon - \eta, \epsilon + \eta))$, $F(\psi_{s'}(\mathbf{y}, t), \mathbf{y}, t) = f(s', \mathbf{y}, t) - \psi_{s'}(\mathbf{y}, t) = 0$. Therefore since $h$ is unique and $f$ is an injection for a fixed $s'$ we have $h = \psi_{s'}$, which means $\psi_{s'}^{-1}$ is continuously differentiable in its domain By Theorem B.1.

We can similarly construct these diffeomorphisms from $\mathbb{R}^{d_a+1}$ to $U \subset \mathcal{S}_e$ for all $s \in \mathcal{S}_e$. Finally, by Definition 2.1 we have that $\mathcal{S}_e$ is a $d_a + 1$ dimensional manifold.

$\square$

Now we prove the general case without the assumption of injectivity on $f$. To do so we construct an surjection $f' : \mathcal{S} \times \mathcal{A}' \times \mathbb{R}^+ \to \mathcal{S}$ where $\mathcal{A}' \subset \mathbb{R}^{d'}$ is an open set such that $d' \leq d_a$. We state the two additional assumptions made in Section 3. The first one being that for any policy $\pi \in \Pi$ we do not have any critical points or loops in the differential field defined by $g_\pi$. Formally, we have the following conditions $g_\pi(s) \neq \mathbf{0}$ for all $s \in \mathcal{S}$ and for a fixed policy $\pi$ the solution for the equation $s_t^\pi = s$, if it exists, is unique in $t$. The second one is that if two actions have the same outcome at one state then they have the same outcome at all states. More formally, there exists an $r > 0$ such that for all $0 < \epsilon < r$, if we have $f(s, a_1, \epsilon) = f(s, a_2, \epsilon)$ for some fixed $a_1, a_2 \in \mathcal{A}$ and any one $s \in \mathcal{S}$ then it holds for all $\mathcal{S}$.

Under the assumptions of Section 2.1 and the ones stated above we restate the main theorem below before presenting the proof.

**Theorem 3.2.** *The set of effective states, $\mathcal{S}_e$, is a smooth manifold of dimensionality at most $d_a + 1$.*

*Proof.* With assumption 3 for every state we have a restriction $\mathcal{A}'_s$ that essentially turns $f(s, a, t)$ into a bijection for every $a \in \mathcal{A}'_s$. Now consider a function $f' : \mathcal{S}_e \times \mathcal{A}'_{s'} \times (\epsilon - \eta, \epsilon + \eta)$ which is a restriction of $f$'s domain to $\mathcal{A}'_{s'} \times (0, T)$ for a fixed $s'$. Now for a fixed $s$, we can construct an open neighbourhood using $\psi_{s'} : \mathcal{A}'_{s'} \times (\epsilon - \eta, \epsilon + \eta)$ constructed as in Proposition B.2, $\psi_{s'}(a, t) = f'(s', a, t)$ such that $\psi_{s'}(a, \epsilon) = s$ for some $a \in \mathcal{A}'_{s'}$ and $\epsilon > \eta > 0$ and also $\epsilon + \eta < \tau$, where $\tau$ is as in assumption 2. Now we know that this is a bijection from assumptions 1, 2 and 3 in Section 3.1 as long as we choose $\epsilon$ and $\eta$ small enough to satisfy the requirements for Proposition B.2. The restricted set $\mathcal{A}'_{s'}$ ensures that each action, for a fixed time $t$ and $s$, maps to a unique state. Further, the assumption on no orbits ensures that for a fixed action $a \in \mathcal{A}'_{s'}$ there is a unique solution to $f(s', a, t) = s''$ for a fixed $a, s'$ and $s''$ in $t$. Therefore, we constructed a full rank smooth bijection $\psi_{s'}$ which maps an open set $\mathcal{A}' \times (\epsilon - \eta, \epsilon + \eta)$ to an open subset of $\mathcal{S}_e$ for every $s \in \mathcal{S}_e$. We can now apply the same arguments of Proposition B.2 to argue that there exists a diffeomorphism $\psi_{s'}$ form an open set in $\mathcal{A}'_{s'} \times (\epsilon - \eta, \epsilon + \eta)$ to an open set in $\mathcal{S}_e$. This holds true for all $s$.

Table 1: Transition Variances for MuJoCo Environments

| Environment | $\mathbf{std}(\mathcal{S}_e)$ |
|---|---|
| Cheetah | $9.4 \times 10^{-16}$ |
| Walker2D | $2.7 \times 10^{-15}$ |
| Reacher | $7.8 \times 10^{-16}$ |
| Swimmer | $1.04 \times 10^{-15}$ |

Finally, we conclude by stating that the dimensionality of the manifold is equal to the of $\dim(\mathcal{A}'_{s'} \times (\epsilon - \eta, \epsilon + \eta)) + 1$ which is less than or equal to $d_a + 1$. □

Finally, we prove Corollary 3.3. The proof is fairly straightforward following Definition 2.2 and Definition 3.1. Since the manifold $\mathcal{S}_e$ is defined as the union of all continuous curves from all possible smooth policies we can find a curve such that for any $s \in \mathcal{S}_e$ and $a \in \mathcal{A}$ for some policy $\pi(s) = a$ in the neighborhood of $s$ $\mathcal{S}_e \cap B(s, \epsilon)$ where $B(s, \epsilon)$ is a Euclidean ball of radius $\epsilon > 0$ centered at $s$. This would mean that for this policy, starting at $s_0$, there is a curve $\gamma$ such that for some time $t$ and some interval $\eta > 0$ we have $\gamma(t) = s$ and therefore $\dot{\gamma}(t) = v$ where $v \in T_s\mathcal{S}_e$. This vector $v$ can now be uniquely mapped to the action and therefore can be labeled $v_a$ as in Corollary 3.3. We also note that this vector is unique independent of the choice of $\epsilon$ or $\eta$.

## C    EMPIRICAL VALIDATION OF OUR ASSUMPTIONS

We validate two of our assumptions. We first demonstrate that the MuJoCo environments, for which we present results in Section 5, are deterministic for all practical intents and purposes. Second, we show that the Jacobian of the tranistion function with respect to the action for a fixed time and state is full rank for all of these environments, thereby suggesting that assumption 1 in Section 3 is "partially satisfied".

### C.1    DETERMINISTIC TRANSITIONS IN MUJOCO ENVIRONMENTS

To estimate the stochasticity of the transitions in the environments we estimate the *transition standard deviation*:

$$\text{std}(\mathcal{S}_e) = \mathbb{E}_{s \sim \mathcal{U}[\mathcal{S}_e], a \sim U[\mathcal{A}]} \left[ (s' - \mathbb{E}[s'])^2 | S_t = s, S_{t+1} = s', A_t = a \right],$$

where $\mathcal{U}[\cdot]$ represents the uniform distribution over a set. This captures how much variance is in the transition dynamics. We estimate this value by randomly and uniformly sampling states, $s$, from an agents trajectories over the learning process. We then obtain randomly and uniformly sample actions, $a$, from the set of actions $\mathcal{A}$. Then for a fixed $(s, a)$ we take observe the one step transition that the environment returns: $s'$, a 100 times. For a fixed $s$ we sample 30 such actions $a$. Therefore, we estimate the quantity $\text{std}(\mathcal{S}_e)$ using approximately 6 million samples, for each environment. The results, divided by the standard deviation of states, are presented in Table 1 suggest that for all practical intents and purposes these environments are deterministic.

### C.2    FULL RANK JACOBIAN ASSUMPTION

To validate assumption 1 from Section 3, we empirically estimate the Jacobian. Our assumption states that the function $f(s, a, t)$ is full rank in $[a, t]$ for all $s \in \mathcal{S}$. Since we are working in practical framework the value of $t$ is fixed to be 1, meaning the environment simulates the application of action $a$ for a single unit of time and returns the next state $s'$. Therefore, we are only able to estimate the Jacobian of $f(s, a, 1)$ for the variable $a$, see that $t$ is fixed to 1. To do so, we sample $s \sim \mathcal{U}[\mathcal{S}_e]$, as described in the previous section. We then sample $a \sim \mathcal{U}[A]$, as described in the previous section. Finally we estimate the Jacobian for the variable $a$ and the function $f(s, a, 1)$, see Theorem B.1 for definition of the Jacobian. We do so using the scipy function `approx_fprime`. Finally, once we have the $d_a \times d_s$ Jacobian matrix we estimate the rank of this matrix using the scipy function `estimate_rank`. We tabulate all the results in Table 2. This procedure comes as close as we can to validating assumption 1 in Section 3.

Table 2: Transition Variances for MuJoCo Environments

| Environment | $d_a$ | Rank |
|-------------|-------|------|
| Cheetah | 6 | 6 |
| Walker2D | 6 | 6 |
| Reacher | 2 | 2 |
| Swimmer | 2 | 2 |

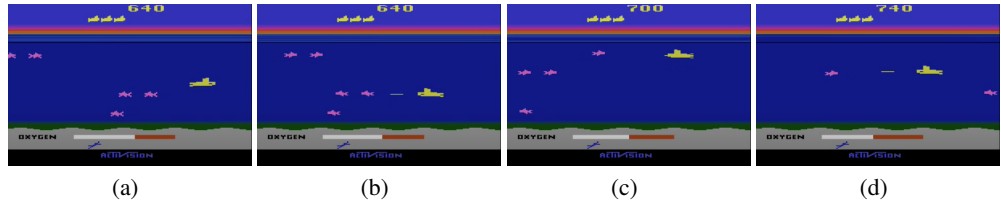

| (a) | (b) | (c) | (d) |

Figure 7: Frames from the Atari game Seaquest. We present 4 frames that are temporally successive from left to right.

## D   CONNECTION TO DISCRETE ACTION ENVIRONMENTS

One of the most popular RL environments with discrete states and actions are the Atari games from the arcade learning environment (Bellemare et al., 2013). Mnih et al. (2013) provided the first result in learning policies with only high-dimensional images as the input to a DNN. Atari games are deterministic given a fixed policy (Hausknecht & Stone, 2015). In case of images, there are two considerations when it comes to the underlying structure of data:

1. **Underlying structure of images:** the states, which are frames from a video game, in Atari games have a low dimensional underlying structure. Even though the frames are $210 \times 160$ pixel images there is a low dimensional underlying structure to them. For example, consider the frames from the Atari game Seaquest in Figure 7. These frames can be identified uniquely with far fewer variables such as: position and heading of the submarine, position and heading of the sharks, oxygen capacity, the scores, the number of lives, position of the bullet and the score.

2. **Underlying structure from the transitions:** Since we know that the agent can only change its state and observation by taking actions it limits the states available to it. In the Seaquest example, we know that the agent can only access some subset of all the possible configurations of the low-dimensional manifold of Seaquest images. Meaning only a subset of configurations of shark positions, submarine positions etc. are realizable.

We further illustrate these ideas in Figure 8. These two factors combined together impose a low-dimensional structure on the state manifold. Development of such a theoretical model, which accurately captures these restrictions or dual structures, is beyond the scope of our current work where we primarily deal with the structure imposed by the transitions.

## E   DIMENSIONALITY ESTIMATION BY FACCO ET AL. (2017)

We describe the algorithm for dimensionality estimation in context of sampled data from the state manifold $\mathcal{S}_e$. Let the dataset be randomly sampled points from a manifold $\mathcal{S}_e$ embedded in $\mathbb{R}^{d_s}$ denoted by $\mathcal{D} = \{s_i\}_{i=1}^N$. For a point $s_i$ from the dataset $\mathcal{D}$ let $\{r_{i,1}, r_{i,2}, r_{i,3}, ...\}$ be a sorted list of distances of other points in the dataset from $s_i$ and they set $r_0 = 0$. Then the ratio of the two nearest neighbors is $\mu_i = r_{i,2}/r_{i,1}$ where $r_{i,1}$ is the distance to the nearest neighbor in $\mathcal{D}$ of $s_i$ and $r_{i,2}$ is the distance to the second nearest neighbor. Facco et al. (2017) show that the logarithm of the probability distribution function of the ratio of the distances to two nearest neighbors is distributed inversely proportional to the degree of the intrinsic dimension of the data and we follow their algorithm for estimating the intrinsic dimensionality. We describe the methodology provided by Facco et al.

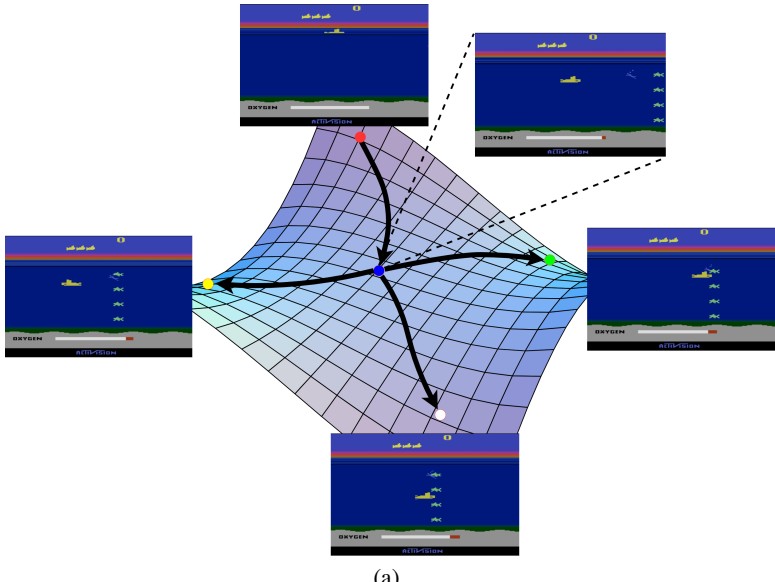

(a)

Figure 8: The structure imposed by transitions is illustrated above. Suppose the agent starts at the red state. If the agent were to execute the down action it gets to the blue state. There are three "directions" in which the agent (and the submarine) can transition. If the agent chooses to execute the action left it gets to the yellow state, similarly green state on right and white state on down. The agent traverses along this surface of admissible images, which are constrained by the underlying structure of the images, given that they are from the Atari game Seaquest and from the set of possible transitions along this manifold. Here discrete actions are represented by continuous curves, in black, on a manifold.

(2017) in context of data sampled by an RL agent from a manifold. Without loss of generality, we assume that $\{s_i\}_{i=1}^N$ are in the ascending order of $r_i$. We then fit a line going through the origin for $\{(\log(\mu_i), -\log(1 - i/N))\}_{i=1}^N$. The slope of this line is then the empirical estimate of $\dim(\mathcal{S}_e)$. We refer the reader to the supplementary material provided by Facco et al. (2017) for the theoretical justification of this estimation technique. The step by step algorithm is restated below.

1. Compute $r_{i,1}$ and $r_{i,2}$ for all data points $i$.

2. Compute the ratio of the two nearest neighbors $\mu_i = r_{i,2}/r_{i,1}$.

3. Without loss of generality, given that all the points in the dataset are sorted in ascending order of $\mu_i$ the empirical measure of cdf is $i/N$.

4. We then get the dataset $\mathcal{D}_{\text{density}} = \{(\log(\mu_i), -\log(1 - i/N))\}$ through which a straight line passing through the origin is fit.

The slope of the line fitted as above is then the estimate of the dimensionality of the manifold.

## F  DDPG Background

An agent trained with the DDPG algorithm learns in the discrete time but with continuous states and actions. With abuse of notation, a discrete time and continuous state and action MDP is defined by the tuple $\mathcal{M} = (\mathcal{S}, \mathcal{A}, P, f_r, s_0, \lambda)$, where $\mathcal{S}, \mathcal{A}, s_0$ and $f_r$ are the state space, action space, start state and reward function as above. The transition function $P : \mathcal{S} \times \mathcal{A} \times \mathcal{S}$ is the transition probability function, such that $P(s, a, s') = \Pr(S_{t+1} = s'|S_t = s, A_t = a)$, is the probability of the agent transitioning from $s$ to $s'$ upon the application of action $a$ for unit time. The policy, in this setting, is stochastic, meaning it defines a probility distribution over the set of actions such that $\pi(s, a) = \Pr(A_t = a|S_t = s)$. The discount factor is also discrete in this setting such that an

analogous state value function is defined as

$$v^\pi(s_t) = \mathbb{E}_{s_l,a_l \sim \pi, P}\left[\sum_{l=t}^{T} \lambda^{l-t} f_r(s_l, a_l)|s_t\right],$$

which is the expected discounted return given that the agent takes action according to the policy $\pi$, transitions according to the discrete dynamics $P$ and $s_t$ is the state the agent is at time $t$. Note that this is a discrete version of the value function defined in Equation 2. The objective then is to maximise $J(\pi) = v^\pi(s_0)$. One abstraction central to learning in this setting is that of the *state-action value function* $Q^\pi : \mathcal{S} \times \mathcal{A} \to \mathbb{R}$, for a policy $\pi$, is defined by:

$$Q^\pi = \mathbb{E}_{s_l,a_l \sim \pi, P}\left[\sum_{l=t}^{T} \lambda^{l-t} f_r(s_l, a_l)|s_t, a_t\right],$$

which is the expected discounted return given that the agent takes action $a_t$ at state $s_t$ and then follows policy $\pi$ for its decision making. An agent, trained using the DDPG algorithm, parametrises the policy and value functions with two deep neural networks. The policy, $\pi : \mathcal{S} \to \mathcal{A}$, is parameterised by a DNN with parameters $\theta^\pi$ and the action value function, $q : \mathcal{S} \times \mathcal{A} \to \mathbb{R}$, is also parameterised by a DNN with ReLU activation with parameters $\theta^Q$. Although, the policy has an additive noise, modeled by an Ornstein-Uhlenbeck process (Uhlenbeck & Ornstein, 1930), for exploration thereby making it stochastic. Lillicrap et al. (2016) optimise the parameters of the $Q$ function, $\theta^Q$, by optimizing for the loss

$$L_Q = \frac{1}{N}\sum_{i=1}^{N}(y_i - Q(s_i, a_i; \theta^Q))^2, \tag{5}$$

where $y_i$ is the target value set as $y_i = r_i + \lambda Q(s'_{i+1}, \pi(s_{i+1}; \theta^\pi); \theta^Q)$. The algorithm updates the parameters $\theta^Q$ by $\theta^Q \leftarrow \theta^Q + \alpha_Q \nabla_{\theta^Q} L_Q$, where $L_Q$ is defined as in Equation 5. The gradient of the policy parameters is defined as

$$\nabla_{\theta^\pi} J(\theta^\pi) = \frac{1}{N}\sum_{i} \nabla_a Q(s, a; \theta^Q)|_{s=s_i, a=\pi(s_i)} \nabla_{\theta^Q} \pi(s; \theta^\pi)|_{s=s_i}, \tag{6}$$

and the parameters $\theta^\pi$ are updated in the direction of increasing this objective.

## G  DDPG MODIFIED ARCHITECTURE COMPARISON

We provide the comparison between single hidden layer network and multiple hidden layer network because our results in section 4 are for single hidden layer. The same architecture is used by Lillicrap et al. (2016) for the policy and value function DNNs which is two hidden layers of width 300 and 400 with ReLU activation. Here we provide the comparison to single hidden layer width 400 and MUP (Yang & Hu, 2020) with GELU activation for the architecture used by Lillicrap et al. (2016). We provide this comparison in Figure 9 and note that the performance remains comparable for both the architectures. All results are averaged over 6 different seeds. We use a PyTorch based implementation for DDPG with modifications for MUP parametrisation and the use of GELU units. The base implementation of the DDPG algorithm can be found here: https://github.com/rail-berkeley/rlkit/blob/master/examples/ddpg.py. The hyperparameters are as in the base implementation.

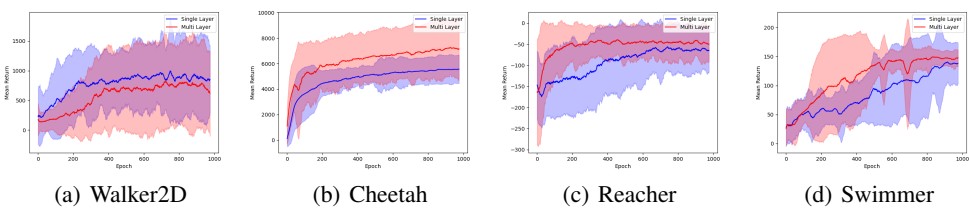

| (a) Walker2D | (b) Cheetah | (c) Reacher | (d) Swimmer |

Figure 9: Comparison of single hidden layer (blue) and multiple hidden layer (red) architectures for DNNs.

## H   Sampling Strategy and Geodesic Distance Estimation

We use graphs as a discrete abstraction of manifolds to sample efficiently whilst preserving *local properties* of the sampled nodes. In many representation learning aprroaches the main objective is to learn to map individual data points to dense vectors in a low dimensional space via stochastic gradient descent (Mikolov et al., 2013; Perozzi et al., 2014; Grover & Leskovec, 2016). One of the primary objectives is to preserve relationships between data points. Graphs, with individual data points mapping to nodes and the relationships between them ecnoded as the edges, are used as an abstraction to represent the whole dataset with all the charecteristic relationships. In our case we seek to preserve *local isometry* in the learnt mapping, $\psi$. Meaning, given a point $s \in \mathcal{D} \subset \mathcal{S}_e$ we would like to preserve the geodesic distance between $s$ and the points in its "neighbourhood" upon being mapped to a lower dimensional representation.

We first define the graph that is a discrete representation the state manifold, $\mathcal{S}_e$, and state the above objective more formally. An undirected graph $\mathcal{G} = (V, E)$ consists of nodes and edges, where $V = \{v_i\}_{i=1}^{N_V}$ are the individual nodes and an edge $e = (v_i, v_j) \in E$ implies that there is a an edge between the nodes $v_i$ and $v_j$ for some $i, j$. Since the graph, $\mathcal{G}$, is undirected $(v_i, v_j) \in E \implies (v_j, v_i) \in E$. Given a node, $v_i$, the one-hop neighborhood of this node is defined as all the nodes $v_j \in V$ such that $(v_i, v_j) \in E$. Now we define a graph, deonted by $\mathcal{G}_\mathcal{D} = (\mathcal{D}, E_\mathcal{D})$ that forms a discrete abstraction over the manifold $\mathcal{S}_e$ using the dataset $\mathcal{D}$. We first set all the ndoes to be the states in our dataset, $\mathcal{D}$. A datapoint's one hop neighbourhood as the $k_{\mathrm{nn}}$ points nearest by the Euclidean distance in the dataset $\mathcal{D}$. This means that for every state $s_i \in \mathcal{D}$ we have $k_{\mathrm{nn}}$ edges to $k_{\mathrm{nn}}$ distinct nearest points in $\mathcal{D}$. This is a $k_{\mathrm{nn}}$-nearest neighbors construction of the graph from a dataset $\mathcal{D}$, sampled from a manifold (Yan et al., 2007; Dong et al., 2011). We augment the edges of this graph $\mathcal{G}_\mathcal{D}$ with an additional set of edge attributes, the Euclidean distance between two points $(s_i, s_j) \in E_\mathcal{D}$ i.e. $||s_i - s_j||_2$. Therefore, the augmented graph now becomes $\mathcal{G}_\mathcal{D} = (\mathcal{D}, E_\mathcal{D}, A_\mathcal{D})$ where $A_\mathcal{D}$ is an ordered set with the Euclidean distances between the two nodes in $E_\mathcal{D}$.

For most practical applications, the number nodes of this graph $\mathcal{G}_\mathcal{D}$ is in thousands and consequently the number of edges are $k_{\mathrm{nn}}$ times the number of nodes. Since we learn the mapping $\psi$ using SGD we can sample batches of state pairs and the distances between them. To do so we randomly sample a subset of states from $\mathcal{D}$. We then construct a subgraph which consists of nodes $k_h$ hops away from these randomly subsampled points. We then use this randomly sampled subgraph to compute the pairwise distance between them using breadth first search. We denote this procedure by the function Random-K-Hop-Subgraph($\mathcal{G}_D, k_h$). Since this randomly sub-sampled graph is much smaller in size whilst preserving the distances we postulate that the distance thus obtained between pairs of states, as the sum of the distance attributes of the shortest paths, is a good approximation of the geodesic distance $d_{\mathcal{S}_e}$. We remind the reader that this geodesic distance is needed for the manifold loss introduced in Equation 3. This three step procedure: **1)** sampling data on the manifold, **2)** constructing a $k_{\mathrm{nn}}$-nearest neighbors graph from this data, and **3)** randomly sampling $k_h$-hop subgraphs, is illustrated with an example of the Swiss roll manifold in Figure 10.

## I   Algorithmic Details

An RL agent, trained using the DDPG algorihm, collects data from trajectories as tuples: $(s, a, s', r)$ where $s$ is the state at which the agent takes an action $a$ and transitions to state $s'$ and obtains the reward $r$. The algorithm stores a buffer of these tuples and therefore we have access to the set of available states $s$, sampled from the state manifold $\mathcal{S}_e$. As described in the previous section, our algorithms requires a dataset sampled from the manifold to estimate the manifold loss in Equation 3. Therefore our algorithm subsamples the states from this buffer to obtain the aforementioned dataset, $\mathcal{D}$, at every episode. Then our algorithm (Algorithm 1) uses this dataset to obtain a sample of pairs of states and geodesic distances between these states, as described in the previous section, to perform SGD on the parameters of the mapping to the low-dimensional space, $\theta_\psi$, on the loss $L_\psi$. All the results have the same values for $k_{\mathrm{nn}} = 6, k_h = 4$ and we use a batch size of 128 for the subsampling the graphs as described in Appendix H for the function Random-K-Hop-Subgraph. These were obtained after manual tuning. The other parameters, are set to the defualt ones for DDPG $\tau = 0.01$, $\lambda = 0.99$, $\alpha_Q = 0.001$ and the replay buffer size is $10^6$. We provide details for running the code in Appendix O.

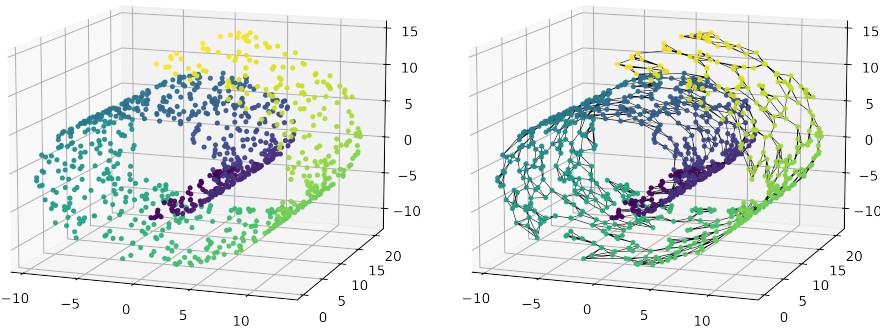

(a) First we obtain a set of points sampled from a manifold.

(b) In the second step, edges are drawn between the $k_{nn} = 4$ nearest points.

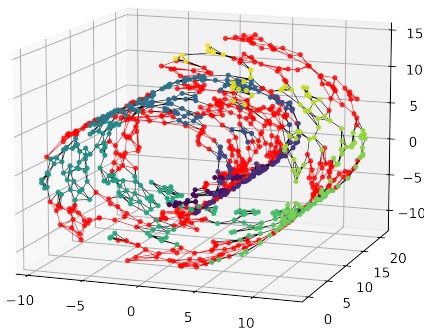

(c) Finally, nodes and edges (in red) are subsampled based on the $k_h = 2$ hop neighbourhood of randomly sampled nodes.

Figure 10: The three step sub-sampling for estimating geodesic distances and batched learning in SGD is illustrated with the example of the Swiss-roll manifold. We use the `scikit-learn` package (Pedregosa et al., 2011) for sampling the data, `torch-geometric` package for generating $k_{nn}$-nearest neighbors graph and also for obtaining the $k_h$-hop subgraphs (Fey & Lenssen, 2019).

---

**Algorithm 1** DDPG with Manifold Representaion Learning

---

$\pi, Q$ parameterised by $\theta^\pi, \theta^Q$ respectively.

$\theta^\pi, \theta^Q \sim \Pr(\theta)$

Initialize the parameters of the target $\theta^{\pi'}, \theta^{Q'} \leftarrow \theta^\pi, \theta^Q$.

Initialise replay buffer $B$

**for** Episode 1 to Max Episodes **do**

    Initialise the Ornstein-Uhlenbeck process for exploration noise: $X$

    **for** Time step $t$, 0 to $T$ **do**

    Set action $a_t = \pi(s_t; \theta^\pi) + X_t$.

    Execute action $a_t$ and observe $r_t, s_{t+1}$ storing it in $B$

    Sample a minibatch of $N$ transitions $(s_i, a_i, r_i, s_{i+1})$ from $B$

    Compute the loss $L_Q$ and update the parameters $\theta^Q$

    Construct the dataset $\mathcal{D}$ from all the states $s_i$ from the sampled $N$ tuples

    Construct the graph $\mathcal{G}_\mathcal{D}$ using $k_{\text{nn}}$-nearest neighbors

    Subsample graph $\mathcal{G}'_{\mathcal{D}'}(\mathcal{D}', E_{\mathcal{D}'}, \mathcal{A}'_{\mathcal{D}'}) = \text{Random-K-Hop-Subgraph}(\mathcal{G}_\mathcal{D}, k_h)$

    Obtain the set of state pairs and geodesic distances, from $\mathcal{G}'_{\mathcal{D}'}$, to calculate $L_\psi$

    Update the policy as in Equation 3:

$$\theta^\pi \leftarrow \theta^\pi + \alpha_\pi \nabla_{\theta^\pi} J(\theta^\pi) - \alpha_\psi \nabla_{\theta^\pi} L_\psi$$

    Update the target networks:

$$\theta^{Q'} \leftarrow \tau\theta^Q + (1 - \tau)\theta^{Q'}$$

$$\theta^{\pi'} \leftarrow \tau\theta^\pi + (1 - \tau)\theta^{\pi'}$$

    **end for**

**end for**

---

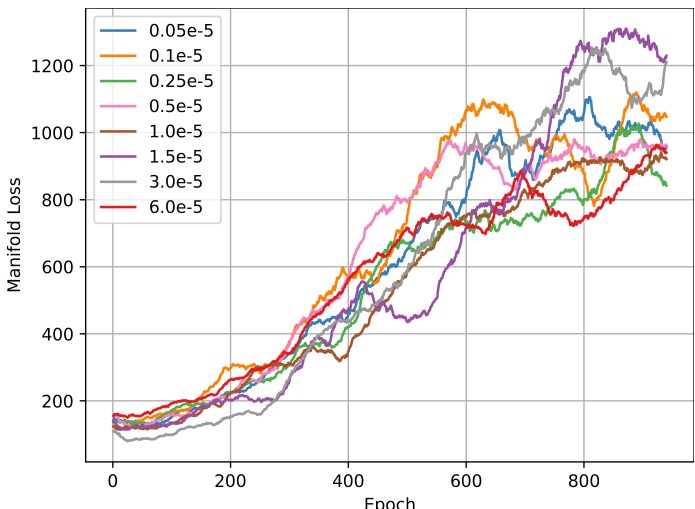

Figure 11: We observe the effect of increasing the hyperparameter $\alpha_\psi$ on the discounted return in the Walker2D environment.

## J ADDITIONAL EXPERIMENTS AND ABLATION STUDIES

All results reported here that are reported are an average over 6 different seeds. For all but the Reacher environment we see the manifold loss decrease as training progresses, as expected. Meaning the agent is able to learn a low-dimensional isometric representation, $\psi$, as well as a policy that operates on this low-dimensional input. We observe that for the Reacher environment our Algorithm is unable

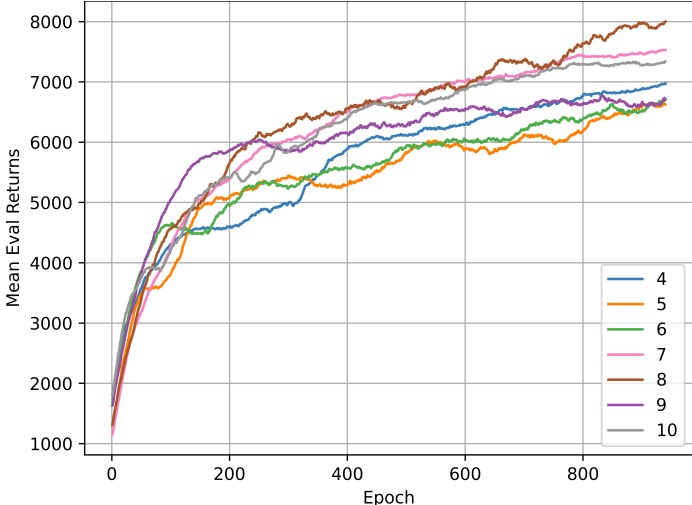

Figure 12: The discounted return for varying the width of the bottleneck layer for the Cheetah domain with $d_a + 1 = 7$. We see the performance peaks at width 8.

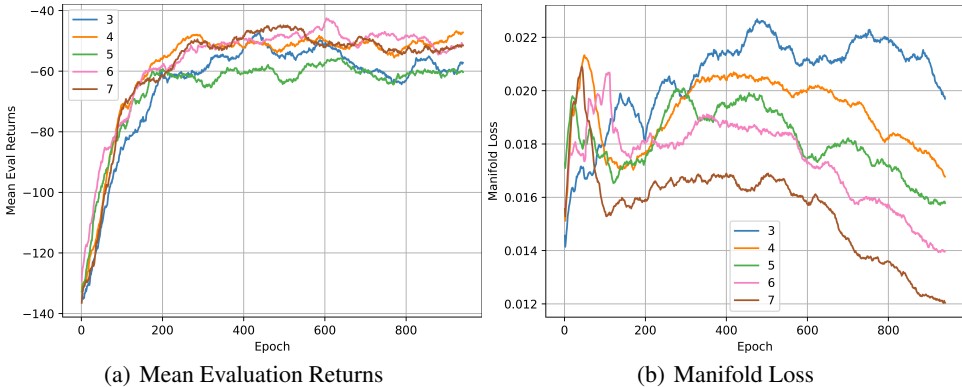

(a) Mean Evaluation Returns  (b) Manifold Loss

Figure 13: We observe that the manifold loss decreases as we increase the width of the bottleneck layer and the performance improves. All the hyper-parmaeters are the same as in Appendix I.

to simulatneously learn a low dimensional mapping and a policy. This is a true despite searching across all the hyperparameters.

We present the ablation study over the hyperparameter $\alpha_\psi$ in Figure 11. This suggests that further increasing the learning rate, $\alpha_\psi$, improves performance of the agent, in case of walker but it begins to detireorate after a certain point, $\alpha_\psi = 1.5$ .

Finally, we demonstrate the effects of changing the width of the bottleneck layer on the Cheetah domain in Figure 12. Most importantly, we observe that the performance peaks at width equal to 8, with both 7 and 8 have similar returns. From the results for dimensionality estimation, in Figure 3, we know that the estimate lies between 7 and 8. This suggests that in case of Cheetah there is an optimal isometric "compression" that allows the agent to perform optimally and better than the baseline. This furthers our argument that an RL agent can learn efficiently on this low dimensional manifold by utilising the underlying structure. The error margins are omitted for the clarity of exposition since there are multiple curves on the same graph.

## K    EXPLAINING FAILURE IN REACHER ENVIRONMENT

Here we explain the failure case of the Reacher environment. As noted in Section 5.2 isometry is a stronger condition that mere diffeomorphism. The Nash embedding theorem states that the an $m$-dimensional $C^1$ manifold can be embedded isometrically into a Euclidean space of dimensionality at most $m(m + 1)$ or $(3m + 11)/2$ (Nash, 1954), therefore the embedding dimension required for learning an isometric embedding for $\mathcal{S}_e$ might be greater than $d_a + 1$. For example, a circle which is a 1D manifold in 2D Euclidean space cannot be isometrically embedded into 1D. Therefore, the objective we train on is stronger than learning a coordinate chart. We hypothesize that for the reacher environment the agent is unable to learn this isometry. As reported in Figure 13, as we increase the width, and therefore the embedding dimension, the manifold loss decreases and the agents performance as measured by mean discounted return improves. As we increase the width to 7 the performance is on par with the baseline approach. The error margins are omitted for the clarity of exposition since there are multiple curves on the same graph.

## L    LEARNING VIA LOW-DIMENSIONAL REPRESENTATION FOR SOFT ACTOR CRITIC

The soft actor critic (SAC) algorithm (Haarnoja et al., 2018) provides a method for learning policy in a more stable manner compared to previous algorithms like DDPG (Lillicrap et al., 2016), TRPO (Bach & Blei, 2015), and PPO (Schulman et al., 2017b).

### L.1    BACKGROUND ON SOFT ACTOR CRITIC

The goal of the SAC algorithm is to train an RL agent acting in the MDP $\mathcal{M} = (\mathcal{S}, \mathcal{A}, P, f_r, s_0, \lambda)$, which is as described in Appendix F. The SAC agent optimises for maximising the modified objective:

$$J(\theta^\pi) = \sum_{t=0}^{T} \mathbb{E}_{s_t, a_t \sim \pi, P} \left[ f_r(s_t, a_t) + \mathcal{H}(\pi(\cdot, s_t; \theta^\pi)) \right],$$

where $\mathcal{H}$ term is the entropy of the policy $\pi$. This additional entropy term improves exploration (Schulman et al., 2017a; Haarnoja et al., 2017). Haarnoja et al. (2018) optimise this objective by learning 4 DNNs: the (soft) state value function $V(s; \theta^V)$, two instances of the (soft) state-action value function: $Q(s_1, a_t; \theta_i^Q)$ where $i \in \{1, 2\}$, and a tractable policy $\pi(s_t, a_t; \theta^\pi)$. To do so they maintain a dataset $\mathcal{D}$ os state-action-reward-state tuples: $\mathcal{D} = \{(s_i, a_i, r_i, s_i')\}$. The soft value function is trained to minimize the following squared residual error,

$$J_V(\theta^V) = \mathbb{E}_{s \sim \mathcal{D}} \left[ \frac{1}{2} \left( V(s; \theta^V) - \mathbb{E}_{a \sim \pi} \left[ Q(s, a; \theta^Q) - \log \pi(s, a; \theta^\pi) \right] \right)^2 \right], \qquad (7)$$

where the minimum of the values from the two value functions $Q_i$ is taken to empirically estimate this expectation. The soft $Q$-function parameters can be trained to minimize the soft Bellman residual

$$J_Q(\theta^Q) = \mathbb{E}_{s, a, r, s' \sim \mathcal{D}} \left[ \frac{1}{2} \left( Q(s, a; \theta^Q) - r - \lambda V(s'; \bar{\theta}^V) \right)^2 \right], \qquad (8)$$

where $\bar{\theta}^V$ are the parameters of the target value function, which are updated at a slower rate compared to the parameters $\theta^V$, as is also done in Algorithm 1. The policy parameters are learned by minimizing the expected KL-divergence,

$$J(\theta^\pi) = \mathbb{E}_{s \sim \mathcal{D}} \left[ D_{KL} \left( \pi(s, \cdot; \theta^\pi), \frac{\exp(Q(s, \cdot; \theta^Q))}{Z_{\theta^Q}(s)} \right) \right], \qquad (9)$$

where $Z_{\theta^Q}(s)$ normalizes the distribution. In addition to this we add the manifold loss as described in Equation 3 to the policy objective. In keeping with the notation above, the learning rates for the functions $V, Q, \pi$ and $\psi$ are $\alpha_V, \alpha_Q, \alpha_\pi$ and $\alpha_\psi$ respectively. Algorithm 2 details how the agent performs this modified learning.

---

**Algorithm 2** SAC with Manifold Representation Learning

---

$\pi, Q_i, V$ parameterised by $\theta^\pi, \theta_i^Q, \theta^V$ respectively.

$\theta^\pi, \theta_i^Q, \theta^V \sim \Pr(\theta)$

Initialize the parameters of the target $\bar{\theta}^V \leftarrow \theta^V$.

Initialise empty dataset $\mathcal{D}$

**for** Episode 1 to Max Episodes **do**

    **for** Time step $t$, 0 to $T$ **do** Sample action $a_t \sim \pi(s_t; \theta^\pi)$.

      Observe successive state $s_{t+1} \sim P(s_{t+1}|s_t, a_t)$.

      Append to dataset $\mathcal{D} \leftarrow \mathcal{D} \cup \{(s_t, a_t, r_t, s_{t+1})\}$.

    **end for**

    Construct the graph $\mathcal{G}_\mathcal{D}$ using $k_\text{nn}$-nearest neighbors

    Subsample graph $\mathcal{G}'_{\mathcal{D}'}(\mathcal{D}', E_{\mathcal{D}'}, \mathcal{A}'_{\mathcal{D}'}) = \text{Random-K-Hop-Subgraph}(\mathcal{G}_\mathcal{D}, k_h)$

    Obtain the set of state pairs and geodesic distances, from $\mathcal{G}'_{\mathcal{D}'}$, to calculate $L_\psi$

    **for** Gradient steps **do**

    Update the value function parameters:

$$\theta^V \leftarrow \theta^V - \alpha_V \nabla_{\theta^V} J_V(\theta^V),$$

    where $J_V$ is as in Equation 7.

    Update the $Q$-function parameters:

$$\theta_i^Q \leftarrow \theta_i^Q - \alpha_Q \nabla J_Q(\theta^Q), i \in \{1, 2\},$$

    where $J_Q$ is as in Equation 8.

    Update the policy following the objective in Equation 9:

$$\theta^\pi \leftarrow \theta^\pi + \alpha_\pi \nabla_{\theta^\pi} J(\theta^\pi) - \alpha_\psi \nabla_{\theta^\pi} L_\psi.$$

    Update the value function target network:

$$\bar{\theta}^V \leftarrow \tau \theta^V + (1 - \tau)\bar{\theta}^V.$$

    **end for**

**end for**

---

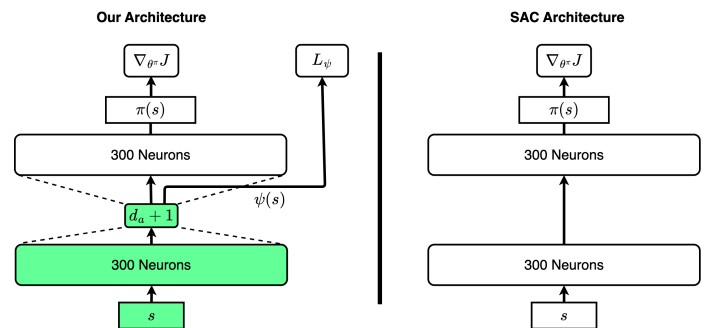

Figure 14: Our architecture for SAC, which similar to the architecture in Figure 4.

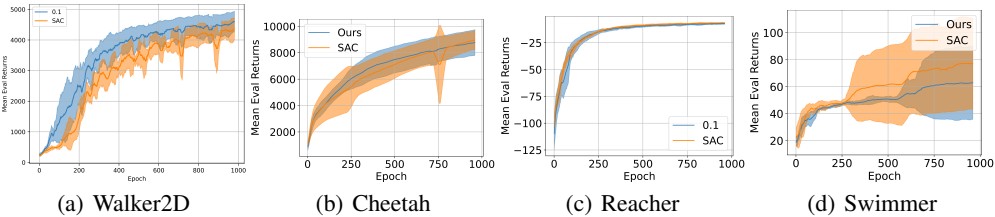

| (a) Walker2D | (b) Cheetah | (c) Reacher | (d) Swimmer |

Figure 15: For all the environments we use $\alpha_\psi = 7.5 \times 10^{-5}$, in comparison to $\alpha_\pi = 3 \times 10^{-4}$, and the rest of the hyper-parameters are the same as reported by Haarnoja et al. (2018), over 6 random seeds.

### L.2 HYPER-PARAMETER DETAILS

The discount factor, $\lambda$, is set to 0.99. The learning rates for the $Q$-functions, $\alpha_Q$, the $V$-function, $\alpha_V$, and the policy, $\alpha_{pi}$ are set to $3 \times 10^{-4}$. The update parameter of the value function, $\tau$, is set to $5 \times 10^{-3}$. The batch size is 256. The learning rate for the manifold loss, $\alpha_\psi$, is set to $7.5 \times 10^{-5}$. All the parameters and for learning the manifold representation are the same as described in Appendix I.

### L.3 EXPERIMENTAL RESULTS

We provide the results for our architecture (as described in Figure 14), with manifold learning, in comparison to "vanilla" SAC for the four algorithms. These results are for an architecture and algorithm similar to described Section 5.2 except with SAC algorithm as opposed to DDPG. All the mean discounted rewards are reported in Figure 15 and the corresponding manifold loss is reported in Figure 16. we observe that our agent performs at par with the baseline in case of Walker2D, Cheetah and Reacher and slightly worse in the Swimmer environment. This result was obtained without an extensive hyperparameter tuning by varying the value of $\alpha_\psi$ over a very small range.

## M COMPARISON WITHOUT MANIFOLD LOSS

We provide another comparison where we compare the same bottleneck architecture for the policy network with and without the manifold loss. This demonstrates the efficacy of the manifold loss

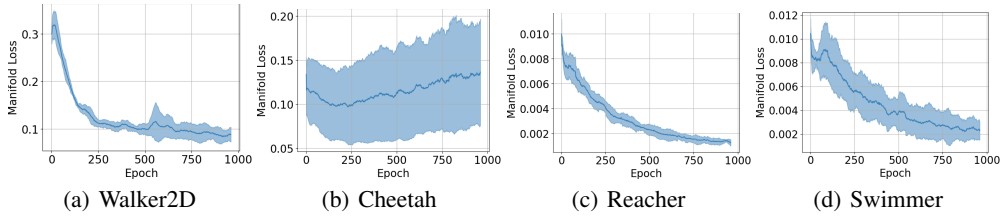

| (a) Walker2D | (b) Cheetah | (c) Reacher | (d) Swimmer |

Figure 16: We report the manifold loss as above. Except Cheetah every other domain behaves as expected.

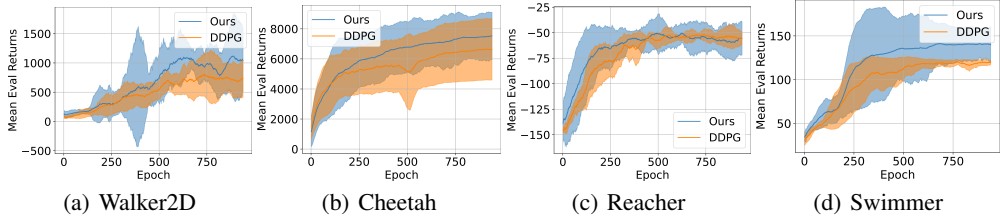

Figure 17: We compare the scenario where we use the same architecture for DDPG as illustrated on the left side of Figure 4, with a bottleneck layer of width $d_a + 1$, with, labeled "DDPG", and without, labeled "Ours", the manifold learning loss. All the results are averaged over 6 random seeds.

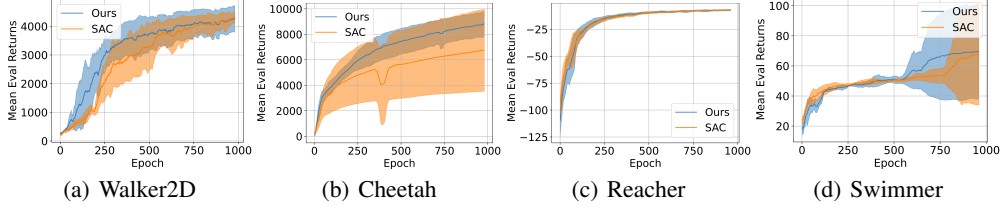

Figure 18: We compare using the same architecture, as in Figure 14, for the two implentations of the SAC algorithm with (labeled "Ours") and without manifold loss (labeled "SAC").

described in Equation 3. In Figure 17 we observe that for all the environments our algorithm performs better except for the Reacher environment where both the algorithms perform sub-optimally to the wide network baseline. We attribute the higher variance to changing representation of $\psi$. Ansuini et al. (2019) showed that DNNs implicitly learn low-dimensional manifold structure at varying depths when trained with SGD in a supervised manner. Their results are for various popular image classification models like ResNet (He et al., 2016), AlexNet (Krizhevsky et al., 2012) and VGG-16 (Simonyan & Zisserman, 2015). This speaks to the efficacy of our manifold loss $L_\psi$, that the addition of this loss improves performance over the implicit low-dimensional manifold representation learnt by a Deep ReLu network with a bottleneck layer. We report a similar comparison between SAC algorithms with the same DNN architecture, as in Figure 14, with and without the manifold loss. We observe that with the manifold loss the performance is better for Walker2D, Swimmer and Cheetah environments. For the Reacher environment we observe that SAC algorithm ends up finding the optimal policy with low variance and with far fewer episodes in both cases.

## N    COMPUTE REQUIREMENTS

For each one of our experiments we perform 6 different runs with varying seeds to obtain the results. Since experiments in Section 5.1 only require trajectories we reuse the trajectories sampled form the ReLU and GELU comparison experiments in Appendix E. For all the experiments in Appendix E we required approximately 180 hours of processing time on NVIDIA GeForce RTX 3090 GPUs which have 24 GB of RAM. We also run two runs of the DDPG algorithm simultaneously on each GPU, owing to the 24 GB RAM availability per GPU, which cuts our run time into half.

## O    STEPS FOR RUNNING THE CODE AND GPU HOURS

All code is in python and was tested for python 3.6. We use the pytorch library for the ease of defining and training DNNs (Paszke et al., 2019) and pytorch_geometric (Fey & Lenssen, 2019) for all graph operations.

We provide the implementation of DDPG using GELU units in the `code/rlkit/` folder of our supplementary material To install the environment we recommend installation using the `setupy.py` file present in the folder. For running the experiment, there are multiple files in the `code/rlkit/examples/smooth_ddpg/ddpg*.py`, each one uses a different architec-

ture and environment details of which can be found within the file. To execute the code, e.g. for `ddpg_arch_2.py`, run the following command from the rlkit folder:

```
python examples/smooth_ddpg/ddpg_arch_2.py 115 gelu
```

where "115" is the seed and "gelu" argument means the DNN thus instantiated uses gelu activations. Our code samples trajectories and the location of the folder can be specified in the file `path_collector.py`

The code for dimensionality estimation using neighborhood data is fairly simple and only requires version 1.1.1 of the python package `scikit-learn` (Pedregosa et al., 2011). It is provided in `code/data-processing/dim-estimate`. The main file is `dim-estimate.py`. It expects a pickle file which is an array of all states sampled for an agent. The code for cleaning the samples, from runs of the DDPG code as described above, and acquiring them in the required format can be found in the file `run_data_processing.py`.

The code for the simultaneous dimensionality reduction and policy learning can be found in the folder `code/rlkit/examples/manifold_learning/ddpg*.py`. The implementation for graph creation and sampling procedure using pytorch_geometric is in the file `code/rlkit/rlkit/torch/manifold/mrl_ddpg.py`. The various architectures used for the policy network can be found in the folder `code/rlkit/rlkit/archs_dir/manfiold_arch`.

Finally, the code for obtaining the illustrations in Figure 10 are in the folder `code/rlkit/illustration_scripts`.

For the results and all its ablations in Section 5.2, we ran multiple instances of the modified DDPG algorithm and the baseline DDPG algorithms for 6 seeds each for 1000 epochs on cloud instances with Nvidia 3090 Ti GPUs with 24 GB memory, and CPUs with 8 cores and 16 GB RAM. Each run takes about 3 hours each. This means we utilised about 700 GPU hours, including hyperparameter tuning and auxiliary experiments presented in the Appendix. The results in Section 5.1 were obtained from the trajectories sampled from various DDPG runs and took about 20 CPU hours on an 8 core machine.

