# OpenReview forum: "On the Geometry of Reinforcement Learning in Continuous State and Action Spaces"
_ICLR.cc/2023/Conference — Submitted to ICLR 2023_

### Official Review · Reviewer_cmmy · 2022-10-20

**Confidence:** 2
**Correctness:** 4
**Technical Novelty And Significance:** 4
**Empirical Novelty And Significance:** 2
**Recommendation:** 6

**Clarity, Quality, Novelty And Reproducibility:**

## Quality + Originality

As stated earlier - the authors present a theoretical contribution for a dimensionality reduction of the effective state space dimension - an assumption that is typically made in prior work.  The empirical results are not convincing, and potential the authors can include a more nuanced discussion on their performance (potentially manifold learning allows for fewer parameters as well, resulting in computational improvements as well).

## Clarity

The submission is well-written and easy to follow. The added discussion and backgrounds on manifolds is appreciated - especially for those like myself who aren't directly inside of this field.  Couple comments for improvements:
- Some of the discussion in the beginning of sections 2.1 and 3 felt repetitive and could be clarified more
- The reduction from the discrete time to the continuous time example was a bit unclear and could be described in more detail in the revision
- More discussion on prior work - what assumptions were made on dimension of the state space, etc


**Strength And Weaknesses:**

## Strengths
1. Model + Theoretical Results: The theoretical results presented in section 3 is novel to the reinforcement learning community.  In fact, this low-dimensional manifold assumption has been made in several prior works - and so this paper can be seen as a proof for the assumptions of those results (although the paper could have explicitly included what those are).
2. Quality of Results: The theoretical results presented in section 3 help highlight the fact that the effective size of solving an MDP problem can be related to the dimension of the action space.  The neural network architecture is novel - combining manifold learning alongside the typical DDPG losses.
3. Relation to Existing Literature: The authors do a good job relating the current analysis to the current literature on manifold learning (highlighting that it has been tested empirically in the past), and neural network architecture design for reinforcement learning.

## Weaknesses
1. Empirical Results: The empirical results show that the performance of the two network architectures (original one + one which uses manifold learning) are statistically similar (due to overlapping confidence intervals).  Potentially there are more extensive empirical results in prior work that could be cited as well for further justification.
2. Theoretical Benefits: The authors should include more theoretical justification of what Theorem 3.2 allows for current RL convergence results.  For example, taking typical function approximation algorithms in the theoretical reinforcement learning community - does knowledge of reduced dimension of $S_e$ versus the state space allow for reduced regret guarantees?  In isolation, the writing and results seem to suggest this fact but the connection is not formally made.

**Summary Of The Paper:**

Reinforcement learning has led to empirical success in complex tasks with continuous state-action spaces in the contexts of game-based and robotics simulations.  Theoretically current tools seem to suggest poor performance - due to the "exponential" scale of the state space representation, even when using strong function approximators like a neural network which is used in practice.  To some extent - these empirical successes have been justified due to the so-called "manifold hypothesis" - that most high-dimensional real-world datasets actually lie on low-dimensional manifolds.  For example, the set of all possible pixel representations of an image is clearly much later than the set of "feasible" images.  Representing the "feasible" region is a low-dimensional manifold, and highlighting that the theoretical guarantees should scale with respect to this manifold instead of the latent space, could be seen as some justification for the results along this lens.  At a high level, this paper essentially proves the manifold hypothesis - stating that the "feasible" state space (i.e. set of states reachable by a smooth policy) has dimension at most the dimension of the action space plus one.  This is applicable for robotics tasks where the state space is the space of all possible images (e.g. 128 x 128 pixels) whereas the actions is a simple 4 dimensional manifold for controlling torque of the four joints.

To be more concrete, the authors consider a continuous-time reinforcement learning with deterministic transitions.  The MDP is represented as $(S, A, f, f_r, s_0, \lambda)$ where $S$ and $A$ are $d_S$ and $d_A$ dimensional subspaces, $f$ is the transition function, $f_r$ the reward function, and $\lambda$ the exponential time-discount factor.  The agent is tasked with learning a policy $\pi$ which is a smooth mapping from states to actions with the goal of maximizing their discounted return, now an integral over time of $f_r(s_t)$ with respect to $t$ where $s_t$ evolves along a curve dictated by the smooth policy.

Under this notation - the authors then define $S_e = \cup_{\pi \in \Pi} ( s | s = H_{\pi}(t) \text{ for some } t )$ as the set of effective states, where $H_\pi$ is the continuous time trajectory or curve of policy $\pi$ in the state space.  Intuitively, this set can be thought of as the set of "reachable" states in this continuous time MDP model.  Note that any state $s$ which is reachable by "some" policy $\pi$ in the policy class $\Pi$ then belongs in $S_e$.  With this, the main result is the following:

**Theorem 3.2** $S_e$ is a smooth manifold of dimension at most $d_a + 1$.

The authors then complement this theoretical discussion with empirical results.  They compare DDPG on four MuJoCo tasks with two network architectures.  The first is a standard one, first discussed in the DDPG paper. The second is a novel architecture which has a bottleneck output of $d_a + 1$ neurons (for learning the manifold) and the rest is a standard layer as in the original architecture. They introduce a modified loss (to ensure that the output serves as a manifold).  The empirical results show similar performance on the original DDPG algorithm and the one with the modified network architecture.

## Questions
- Do prior works which consider manifold learning on top of developing an RL algorithm essentially just tune the resulting manifold dimension?
- Discussion on top of page 6 is confusing - state that you can find such a $v$ but then state that the existence is beyond the scope?

## Minor Comments
- Last sentence of introduction is unclear and could be rewritten and clarified.
- "This solution" on top of page 3 - solution to what?
- Superscript of $\pi$ missing in $s_l$ in Equation 3
- First paragraph of section 2.2 is a bit unclear
- "start" instead of "starting" in first paragraph of section 3
- Restate definition of $H_\pi$ in Definition 3.1
- Theorem ?? on page 7
- Plots are hard to read in grey-scale



**Summary Of The Review:**

The authors include strong theoretical contributions on state-space reduction.  However, the empirical results are not convincing, and there are not any concrete theoretical justification on the benefits of their main result (Theorem 3.2).

---

> ### Author Response · Authors · 2022-11-09
> **Response to Reviewer cmmy**
>
> Thank you very much for your thorough reading of our work and the detailed review. We answer all of your queries below.
>
> >*Empirical Results: The empirical results show that the performance of the two network architectures (original one + one which uses manifold learning) are statistically similar (due to overlapping confidence intervals). Potentially there are more extensive empirical results in prior work that could be cited as well for further justification.*
>
> We have added context from previous empirical work and mention how this suggests that manifold learning based approaches are a promising direction for improving the performance of RL agents. Thank you for pointing this out.
>
> >*Theoretical Benefits: The authors should include more theoretical justification of what Theorem 3.2 allows for current RL convergence results. For example, taking typical function approximation algorithms in the theoretical reinforcement learning community - does knowledge of reduced dimension of  versus the state space allow for reduced regret guarantees? In isolation, the writing and results seem to suggest this fact but the connection is not formally made.*
>
> Thank you for pointing out this was missing from our work. We have now, in the latest version, added a paragraph in the related works section on how work can impact various theoretical bounds. We cite multiple papers in deep learning theory [1,2,3,4] that show that the sample complexity and learning dynamics depend strongly on the intrinsic dimension and weakly on the ambient dimension. More importantly, we note that [5] have shown that the sample complexity of off-policy evaluation depends strongly on the intrinsic dimensionality of the manifold and weakly on the embedding dimension. Our result gives an upper bound on the manifold dimension and therefore coupled with [5] it suggests that the sample complexity is upper bound by an exponential function of $d_a +1$ and not $d_s$. Making a formal and precise connection will require rigorous consideration which is beyond the scope of current work. We hope that this answers your concern.
>
> >*Some of the discussion in the beginning of sections 2.1 and 3 felt repetitive and could be clarified more*
>
> We have abridged the discussion at the start of Section 3. Thank you for pointing out that we were being repetitive with the example of Cheetah.
>
> >*The reduction from the discrete time to the continuous time example was a bit unclear and could be described in more detail in the revision*
>
> We have now rewritten the discrete time to continuous time scenario. We argue that since there exists a discretisation of the continuous physical time process we can find an inverse: namely the $\vartheta$ that smoothly transforms action $a_t$ to $a_{t + 1}$. Please let us know if it is clearer now.
>
> >* More discussion on prior work - what assumptions were made on dimension of the state space, etc*
>
> Unfortunately, in previous work there are no explicit assumptions on the dimensionality or existence of the manifold other than implicit assumptions that it exists and is lower (or far lower) dimensional. We have been unable to find any such explicit statements but we’d be happy to include them if you are aware of any.
>
> **References:**
>
> [1] Uri Shaham, Alexander Cloninger, and Ronald R. Coifman. Provable approximation properties for deep neural networks
>
> [2] Alexander Cloninger and Timo Klock. Relu nets adapt to intrinsic dimensionality beyond the target domain
>
> [3] Sebastian Goldt, Marc Mézard, Florent Krzakala, and Lenka Zdeborová. Modelling the influence of data structure on learning in neural networks.
>
> [4] Sam Buchanan, Dar Gilboa, and John Wright. Deep networks and the multiple manifold problem.
>
> [5] Xiang Ji, Minshuo Chen, Mengdi Wang, and Tuo Zhao. Sample complexity of nonparametric off-policy evaluation on low-dimensional manifolds using deep networks.

---

> > ### Comment · Reviewer_cmmy · 2022-11-19
> > **Response**
> >
> > Thanks for addressing all of my comments.  I enjoy the new discussion around the literature, especially citing the papers in deep learning theory highlighting how performance scales with the intrinsic dimension of the problem.

---

### Official Review · Reviewer_RjLF · 2022-10-24

**Confidence:** 4
**Correctness:** 3
**Technical Novelty And Significance:** 3
**Empirical Novelty And Significance:** 4
**Recommendation:** 8

**Clarity, Quality, Novelty And Reproducibility:**


- The proposed analysis in the paper focuses on mujoco environments that are deterministic and in the continuous state in action space. What are the connections between the methods described in this work and discreet action spaces such as the Atari environments? Does the analysis in the paper help us determine the intrinsic state space in image-based worlds or worlds with discrete actions?
- One of the cons of this method is that it could be an indicator that the assumptions that the method is continuous and deterministic also imply that this method only works in very simple environments. This could speak more to the non-realistic assumptions which this type of method can be applied. In this type of environment, it could be possible to train a policy where the state space only consists of the time $t$. If the proposed analysis can be generalized to complex and more general environments, it'll be more helpful to the community.
- Also, the assumption of no orbits in theory, which can also be interpreted as the probabilistic graph for the policy must be directed and not have cycles, is also a very strong assumption and that the agent will never revisit a state that visited at a prior time.
- In the empirical analysis section of 3.2, how is a low dimensional mapping learned from the state space to the action space plus one? The details around how this learning and compression algorithm is designed are important to understanding the application of the method for this experiment.
- In order to train the coordinate chart, is supervised information needed that dictates exactly what the distance between two states should be in the state space? It is not clear where the supervised information comes from next on the left side of equation 3.
- How many random scenes were used for figure 5? It is not clear what implications can be made from figure 5. Is this a proof of concept?

**Strength And Weaknesses:**

pros
- The first pro for this method is that it does appear to have a rather helpful proof in showing how the dimensions for a control problem are also heavily connected to the action space for the problem.
- The analysis also includes two different types of empirical experiments to better understand how close do the theoretical analysis is to a practical comparison.

cons
- The analysis, including the theory in the paper, is limited to a very small subset of possible planning problems. While the analysis may be correct, the fact that it only works on continuous state and action problems that are deterministic might say more about the limited applicability of the intrinsic dimensionality findings.
- This is also reflected in some of the experiments that analysis or it would be helpful to include analysis where the state space is image-based rather than the already dense information from the robot poses.

**Summary Of The Paper:**

This paper proposes a theoretical analysis to understand the intrinsic dimension for deterministic continuous-time State and action space problems. This method is motivated by understanding the representations required to train optimal policies in these types of spaces will help us better understand the structure of networks that can be used to solve these types of tasks. The method performs some theoretical analysis to indicate that a diffiomorphism can be found between the continuous time optimization problems such that the intrinsic space is only the size of the action space + 1. Two different experiments are shown to try and empirically justify the results that analyze the reproducibility of the theoretical results in some of the mujocu simulated robotic control environments.

**Summary Of The Review:**

The work forms a theoretical analysis to indicate the intrinsic dimension of a continuous time control system. The motivation and scope of the contribution need more details before the score can be raised.

---- Updated score
After more clarity on the method and a few updates to the framework.

---

> ### Author Response · Authors · 2022-11-09
> **Response to Reviewer RjLF**
>
> Thank you very much for your detailed and thoughtful review of our work. We answer all your concerns below.
>
> >*The proposed analysis in the paper focuses on mujoco environments that are deterministic and in the continuous state in action space. What are the connections between the methods described in this work and discreet action spaces such as the Atari environments? Does the analysis in the paper help us determine the intrinsic state space in image-based worlds or worlds with discrete actions?*
>
>
> Atari environments are primarily image inputs. As noted in our work, there is a huge body of work discussing how natural images lie on a low dimensional manifold embedded in a high dimensional space [1, 2, 3]. This could serve as an extension to our argument and one could imagine the agent observing data sampled from this manifold of images. The focus of our work is to introduce, prove, and demonstrate how simulated robotic control tasks have an underlying low dimensional structure which is rooted in the continuous and somewhat smooth nature of the actions themselves. A much more detailed analysis of discrete action and stochastic transition environments is out of the scope of our current study. In the revised version, we do provide a comment on discrete action spaces in Appendix D that serves as a pointer for future research. We hint at a theoretical model which extends discrete actions to be continuous actions, of fixed time interval, with the agent traversing a continuous path along the low-dimensional manifold of images. Since the low-dimensional structure of the state space, in this case, is given by both the manifold of images and the transitions of an agent, it requires careful consideration and rigorous treatment.
>
> >*One of the cons of this method is that it could be an indicator that the assumptions that the method is continuous and deterministic also imply that this method only works in very simple environments. This could speak more to the non-realistic assumptions which this type of method can be applied. In this type of environment, it could be possible to train a policy where the state space only consists of the time . If the proposed analysis can be generalized to complex and more general environments, it'll be more helpful to the community.*
>
>
> **On cons of our method and generalizing to complex environments:** our work is primarily theoretical in nature with rigorous empirical validation. Here, we have ensured the marriage of theory and practice. To that effect we have added the verification of two of our assumptions in Appendix C (to the best of our abilities):
> i) We verify that MuJoco domains are in fact deterministic. This is done by sampling multiple transitions for pairs of state and action and observing the standard deviation in the next state. MuJoCo  domains, to the best of our knowledge, have not been solved using only time $t$. This also suggests that learning a simple policy based on time $t$ might not be feasible, as claimed by you.
>  ii) We have also verified the full rank jacobian assumption from Section 3.2 (assumption 1) for MuJoCo environments. We do not have access to the transition function $f(s, a, t)$, for a fixed s. We do have access to the function $f(s, a, 1)$, meaning the transition function over the application of action $a$ for 1 unit of time at state $s$.We empirically estimate the Jacobian of the function $f^{\text{restricted}}_s(a) = f(s, a, 1)$ across different states $s$. We observe that the rank of this function $f^{\text{restricted}}_s$ is the same as $dim(\mathcal A)$, thereby hinting at the validity of our assumption.
>
> While we do make simplifying assumptions many of them in fact hold in practice! The main result, Theorem 3.2, also holds in practice as shown by our dimensionality estimates in Section 5.1. This furthers our argument that there is a low dimensional underlying structure to RL problems. Moreover, our work opens up avenues for manifold based learning by forming a strong theoretical backbone that is applicable to practice.
>
>
> **References:**
>
> [1] Joshua B. Tenenbaum. Mapping a manifold of perceptual observations. In NIPS, 1997.
>
> [2] Michael M. Bronstein, Joan Bruna, Taco Cohen, and Petar Velivckovi’c. Geometric deep learning: Grids, groups, graphs, geodesics, and gauges. ArXiv, abs/2104.13478, 2021
>
> [3] C. Fefferman, S. Mitter, and Hariharan Narayanan. Testing the manifold hypothesis. arXiv: Statistics Theory, 2013.

---

> > ### Author Response · Authors · 2022-11-09
> > **Response to Reviewer RjLF Continued**
> >
> > >*Also, the assumption of no orbits in theory, which can also be interpreted as the probabilistic graph for the policy must be directed and not have cycles, is also a very strong assumption and that the agent will never revisit a state that visited at a prior time.*
> >
> > We would like to point out that this is in fact incorrect. The assumption is that there exists a $\tau > 0$ such that for all $t < \tau$ there are no orbits. This means that the agent can visit the state again BUT not under time $\tau > 0$. We apologize for the confusion. We have now renamed the assumption to “no small orbits”. Note that this was the assumption all along we just are renaming it in the latest version. This assumption can be interpreted as the MDP not being reversible in infinitesimally small time. This is a difficult assumption to validate but we believe that it is not unrealistic. For example, in the Walker2D environment, intuitively, for the agent to go back to any state it will take a non-zero time and therefore there exists such a $\tau > 0$. It is naturally true in environments with physical dynamics, e.g., acceleration, velocity or displacement cannot be instantaneously reversed.
> >
> > >*In the empirical analysis section of 3.2, how is a low dimensional mapping learned from the state space to the action space plus one? The details around how this learning and compression algorithm is designed are important to understanding the application of the method for this experiment.
> > In order to train the coordinate chart, is supervised information needed that dictates exactly what the distance between two states should be in the state space? It is not clear where the supervised information comes from next on the left side of equation 3.*
> >
> >
> >
> > Assuming you mean section 5.2 above. We had detailed how this geodesic distance in Equation 3, $d_{\mathcal S_e}(s_1, s_2), is calculated and how we sample pairs of states in Appendix F of the original version (and Appendix H of the new version). We have also pointed to the Appendix in the main body. We also refer you to Figure 10 which has an illustration to clarify this further. All of this was done due to space constraints. In order to clarify this further we have now added to the previously brief description of the estimating procedure in Section 5.2, please see the paragraph after Equation 3. In summary it is a three step process:
> > 1. Sample points from the replay buffer and therefore the state manifold
> > 2. Construct a graph using k-nearest neighbors
> > 3. Measure distance between sub-sampled pairs of states using a variant of breadth first search on this graph.
> >
> > These three steps are illustrated in Figure 10 in the Appendix. This is not a supervised learning approach. It is an unsupervised loss based on a graph based distance metric. We estimate this distance to calculate the loss in tandem with policy learning (see Algorithm 1 in the Appendix for more details). Thank you for pointing out this was unclear. We have added details in Section 5.2.
> >
> > >*How many random scenes were used for figure 5? It is not clear what implications can be made from figure 5. Is this a proof of concept?*
> >
> > We use 6 different seeds for each graph, we had earlier reported this in the Appendix but we now do so in the main body of the paper. Thank you for pointing this out. The intent  of these graphs is to show that a DDPG agent can learn a policy effectively with this highly compressed representation. It is a demonstration that this low-dimensional space is sufficient for policy learning. Not only do we show that there is a parallel to the manifold hypothesis in RL, we also show that this low-dimensional structure is applicable for learning a policy!

---

> > > ### Comment · Reviewer_RjLF · 2022-11-24
> > > **Response**
> > >
> > > Thank you for the additional clarity around the method. The updates for the limitations around no orbits are helpful and increase the applicability of the method. There are still limitations to applying this method to any stochastic MDP or POMDP.
> > >
> > > I have updated my score.

---

> > > > ### Author Response · Authors · 2022-11-25
> > > > **Thank you for raising the score**
> > > >
> > > > Thank you very much for reading our response and raising the score.

---

> ### Author Response · Authors · 2022-11-23
> **Any remaining issues or comments?**
>
> Please let us know in case there are any lingering comments or concerns. We have tried to address all your concerns regarding the assumptions and also added experiments supporting these assumptions.

---

### Official Review · Reviewer_R4ss · 2022-10-26

**Confidence:** 3
**Clarity, Quality, Novelty And Reproducibility:** The paper is easy to follow in general.
**Correctness:** 3
**Technical Novelty And Significance:** 2
**Empirical Novelty And Significance:** 2
**Recommendation:** 5

**Strength And Weaknesses:**

Strength
- The paper provides an interesting analysis on the geometry of the state space. For a deterministic environment satisfying the three conditions stated in the paper, the effective/reachable state space is shown to be a low-dimension manifold whose dimension is determined by the dimension of the action space.

- The upper bound on the effective dimension is shown to be close to the empirical estimate, and this effective dimension could potentially help the architecture design of value and policy networks in RL algorithms.

Weaknesses
- The analysis only works for deterministic environments. In fact, with noise in the transition dynamics, the dimension of the effective state space will be the entire state space for a general problem. There is a paragraph briefly discussing the possible extension to stochastic environments with additive Gaussian noise, but no actually analysis for the stochastic case.

- The three assumptions seem pretty restrictive. They are essential for the proof as discussed in Appendix A, but they basically restrict the model to be rather simple and not likely to hold in interesting problems like those MuJoCo environments. Does any of the MuJoCo environment (no randomness) satisfy these assumptions?

- In numerical experiments, only one architecture with a bottleneck of $d_a + 1$ is considered. If the goal is to show that $d_a + 1$ is the effective dimension and is sufficient for state representation in a RL algorithm, one may need to try multiple architectures with different bottleneck dimensions and compare their performance.

**Summary Of The Paper:**

The paper shows that, under certain conditions, the set of reachable states is a smooth manifold with dimension at most the dimension of the action plus one. The paper then proposes a DNN architecture with a bottleneck and show its competitive performance in numerical experiments.

**Summary Of The Review:**

The analysis on the geometry of effective state space is interesting, but the theoretical results only works for a very limited class of problems. Numerical experiments may be a bit limited for validating the low-dimensional effective state space property.

---

> ### Author Response · Authors · 2022-11-09
> **Response to Reviewer R4ss**
>
> Thank you for your valuable comments, appreciation of our analysis and empirical approach to theoretical results. We respond to your concerns below.
>
> >  *The analysis only works for deterministic environments. In fact, with noise in the transition dynamics, the dimension of the effective state space will be the entire state space for a general problem. There is a paragraph briefly discussing the possible extension to stochastic environments with additive Gaussian noise, but no actually analysis for the stochastic case.*
>
> **The analysis only works for deterministic environments:** Mujoco environments with control inputs, which are known to have deterministic transitions, are widely used and challenging environments for benchmarking algorithms. We have added experiments and all the details to confirm this assumption in Appendix C (of the revised version of our paper). We observe that the normalized standard deviation of transitions is of the order $10^{-16}$ thereby validating our deterministic transitions assumption.
>
> While our assumption might seem restrictive at a first glance, we have shown that practical and popular MuJoCo environments used for benchmarking RL algorithms do satisfy it. What is surprising is how these popular environments used in continuous RL are actually deterministic, despite most other works developing practical algorithms for the stochastic case. This new experiment, in addition to our empirical validations in Section 5, speaks to the applicability of our results.
>
> In the stochastic case we postulate that if the noise is small in magnitude then all the data is concentrated around a manifold. This would still mean that the data is not distributed over the entirety of $R^n$ and speaks to the applicability of our results in that case.
>
> Please let us know if this addresses your concern.
>
> > *The three assumptions seem pretty restrictive. They are essential for the proof as discussed in Appendix A, but they basically restrict the model to be rather simple and not likely to hold in interesting problems like those MuJoCo environments. Do any of the MuJoCo environments (no randomness) satisfy these assumptions?*
>
> **On the other three assumptions for Theorem 3.2:** as pointed out by you, these are essential for our proof. We argue that assumptions 2, 3 are not very restrictive and based on a new experiment assumption 1 is valid.
>
> On the “no orbits” assumption 2:  We have now renamed the second assumption to “no small orbits” instead of “no orbits”. We argue that this is not very restrictive in Appendix A of the revised version. In summary, as long as there exists a $\tau > 0$ such that there are no orbits for time $t < \tau$ our assumption is satisfied. For practical environments, we argue that this $\tau$ can be sufficiently small. Consider the Walker2D environment with the walker moving forward with a non-zero speed, suppose the agent is at state $s$ at time $t$ for the agent to go back to $s$ it will take a non-zero time and therefore there exists such a $\tau > 0$.
>
>
> The third assumption, that of an open action restriction set $\mathcal A’$, is much more difficult to verify since it involves solving for the equation $ f(s, a_1, t) = f(s, a_2, t)$ under the condition $a_1 \neq a_2$. This is an assumption about restricting $\mathcal A$ to $\mathcal A’$ such that the transitions are unique. The absence of such a restriction would mean that we cannot choose a subset of actions which make transitions unique which seems unreasonable. To the best of our knowledge, we do not know of any practical methods that solve this equation.
>
> For verifying the full rank assumption we do not have access to the transition function $f(s, a, t)$, for a fixed s. We do have access to the function $f(s, a, 1)$, meaning the transition function over the application of action $a$ for 1 unit of time at state $s$.We empirically estimate the Jacobian of the function $f^{\text{restricted}}_s(a) = f(s, a, 1)$ across different states $s$. We observe that the rank of this function $f^{\text{restricted}}_s$ is the same as $dim(\mathcal A)$, thereby hinting at the validity of our assumption. We have added the details and results of this empirical analysis in Appendix C of the revised version of our paper.
>
> **In summary, we believe that our assumptions are less restrictive than they might appear.** Despite its limitations, our work not only holds true in theory but is also very close to practice, as can be seen from our validation of our theoretical results in Section 5.

---

> > ### Author Response · Authors · 2022-11-09
> > **Response to Reviewer R4ss Continued**
> >
> > >*In numerical experiments, only one architecture with a bottleneck of $d_a + 1$ is considered. If the goal is to show that  is the effective dimension and is sufficient for state representation in a RL algorithm, one may need to try multiple architectures with different bottleneck dimensions and compare their performance.*
> >
> > On the experiment with different bottleneck dimensions and widths: we have provided this experiment in Figure 13 of Appendix J titled “ADDITIONAL EXPERIMENTS AND ABLATION STUDIES” in the latest version (or Appendix H in the first version). In summary, for the Cheetah domain we show that varying bottleneck dimensions in the neighborhood of $d_a + 1 = 7$ we see that the network architecture with bottleneck dimensions 7 and 8 are the best performing. This result was included in our original submission. We have also added a sentence in the main body pointing to this result in Section 5.2. Please let us know if this addresses your concern.

---

> ### Author Response · Authors · 2022-11-23
> **Further comments or concerns?**
>
> We hope we have answered all your concerns regarding the assumptions and our new experiments validating them are sufficient. Please take a look at the latest version of our paper. Please let us know in case there are any concerns.

---

> > ### Author Response · Authors · 2022-12-05
> > **A Gentle Reminder**
> >
> > Thank you very much for raising concerns on the applicability of our method and the assumptions we have made. As detailed in our comment, we have explained better the applicability of our method. Please do take a look at Appendix C where we verify two of our "big" assumptions.

---

> ### Author Response · Authors · 2022-12-13
> **Additional reminder**
>
> Thank you very much for your valid queries on the applicability of our method and the assumptions we have made. Please do go over our comments and the latest revision of our work. Also, see that reviewer RjLF who had similar concerns has raised their score.

---

### Official Review · Reviewer_HpSc · 2022-10-27

**Confidence:** 4
**Correctness:** 3
**Technical Novelty And Significance:** 3
**Empirical Novelty And Significance:** 3
**Recommendation:** 5

**Clarity, Quality, Novelty And Reproducibility:**

$\textbf{Clarity: }$

The writing and presentation of the proposed method is almost clear. The organization of this paper is good.

&nbsp;

$\textbf{Novelty: }$

To my knowledge, the theoretical and expirical results on state manifold in RL is novel.

&nbsp;

$\textbf{Quality: }$

The theoretical part is almost clear yet I also have questions for the authors. The empirical validation is interesting while is insufficient in several aspects as I point out in the part of Strengths and Weaknesses.

&nbsp;


$\textbf{Reproductibility:}$

The proposed algorithm is clear and it seems to be easy to implement. The source codes are also provided.


**Strength And Weaknesses:**

$\textbf{Strengths:}$
+ I appreciate the authors’ efforts in study the manifold of effective states in MDP. Personally, I am interested in it and I think it is fundamental to efficient RL in complex problems.
+ The writing and presentation of the proposed method is almost clear. The organization of this paper is good.
+ The theoretical results and the empirical validation match well to some extent.

&nbsp;

$\textbf{Weaknesses (and Questions): }$

The main theoretical contribution is Theorem 3.2. By referring to Appendix B, I am confused on the proof of Proposition B.2. Concretely, since $\phi_{s^{\prime}}: A \times (\epsilon - \eta, \epsilon + \eta) \rightarrow U \in S_{e}$, the inverse $\phi_{s^{\prime}}^{-1}$ should be a mapping from state to action and time interval, right? This turns to be confusing when the authors derive the equivalence between $h$ and $\phi_{s^{\prime}}^{-1}$. Do I misunderstand some part?



&nbsp;


My other concerns are on the empirical validation.



For the empirical dimensionality estimation, I feel a gap between theory and practice, i.e., the policies (DDPG policies) used to sample state data are poor in performance in MuJoCo and thus should only cover a very small part of all possible smooth policies.

Therefore,  I question the validity and generality of the results of estimated manifold dimensionality.

In addition, what is the exact number of the size $n$ of the sampled states used to estimate the dimensionality in Section 5.1? I am also curious about how different numbers of $n$ affect the dimensionality estimated.

&nbsp;

Similarly, personally, I am not satisfied with the results in Section 5.2, because I am very familiar with the performance of different representative DRL algorithms in MuJoCo and I do not think DDPG is a good baseline algorithm in MuJoCo.

I recommend the authors to conduct the expriments based on TD3 or SAC which will be much more convincing, at least to me.

For the comparison in Figure 5, do the authors have the results of a baseline that keeps the $d_a + 1$ bottleneck structure while not uses $L_{\phi}$ (i.e., train as the original DDPG). I think this baseline will help a lot in understanding the efficacy of $L_{\phi}$.

&nbsp;


For some additional discussion, I wonder the authors’ opinion on how this work, i.e., the manifold of effective state space, relates or differs to recent works on studying low-rank state (representation) space in DRL, e.g., [1,2].

&nbsp;


Reference:

[1] Yuzhe Yang, Guo Zhang, Zhi Xu, Dina Katabi. Harnessing Structures for Value-Based Planning and Reinforcement Learning. ICLR 2020

[2] Aviral Kumar, Rishabh Agarwal, Dibya Ghosh, Sergey Levine. Implicit Under-Parameterization Inhibits Data-Efficient Deep Reinforcement Learning. ICLR 2021

&nbsp;

Minors:
- The reference in the second paragraph in Section 5.2 is broken.
- The figures are in low fidelity. The font size of the texts in the figures are too small to read.



**Summary Of The Paper:**

This paper studies the manifold of effective (or reachable) states in MDP with continous states and continuous actions. Based on the continuous-time modeling, the authors present their main theorical contribution, i.e., the theory that proves the set of effective states is a smooth manifold of dimensionality at most the dimensionality of action space plus 1. Then, the authors provide empirical validation from two aspects: 1) the empirical estimation of the effective state manifold, and 2) the performance evaluation of a DDPG variant equipped a low-dimensional manifold representation by learning from the knn-graph estimated geodesic distance.

**Summary Of The Review:**

According to my detailed review above, I think this paper is below the acceptance threshold mainly due to my concerns on the theory and the empirical results (see concrete content above).

I am willing to raise my rating if my concerns are well addressed.

---

> ### Author Response · Authors · 2022-11-14
> **Response to Reviewer HpSc**
>
> Thank you for your insightful and detailed review. Our apologies for the delay in responding to you, this was because we were running time and compute intensive experiments in lieu of concerns raised by you. We respond to your comments below.
>
> >*The main theoretical contribution is Theorem 3.2. By referring to Appendix B, I am confused on the proof of Proposition B.2. Concretely, since $\phi_{s’}: \mathcal A \times (\epsilon - \eta, epsilon + \eta) \to U \subset \mathcal S_e$, the inverse $\phi_{s’}^{-1}$ should be a mapping from state to action and time interval, right? This turns to be confusing when the authors derive the equivalence between h and $\phi_{s’}^{-1}$. Do I misunderstand some part?*
>
> Thank you very much for pointing out this typo in the proof. The equivalence is between $h$ and $\psi_{s’}$ and by implicit function theorem the inverse of $h$ is differentiable. We have changed the proof and the statement of the theorem to clarify it further. This does not alter any of the assumptions or our proof. Please let us know if the proof is clearer now.
>
> >*For the empirical dimensionality estimation, I feel a gap between theory and practice, i.e., the policies (DDPG policies) used to sample state data are poor in performance in MuJoCo and thus should only cover a very small part of all possible smooth policies. Therefore, I question the validity and generality of the results of estimated manifold dimensionality.*
>
> In general, any learning algorithm covers only a small subset of policies from the set of all policies. Even if we were to use any other algorithm, e.g. SAC, for this sampling process a better performing algorithm might cover an even smaller set of policies by virtue of finding an optimal policy in fewer episodes.  More importantly, our experiments pertain to the “coverage” of states: do the sampled states come from the state manifold? Further note that the algorithm used by us to measure the dimensionality does not rely on a uniform sampling from the manifold [3]; instead it relies on sampling from a distribution with support on the manifold.
>
> > *In addition, what is the exact number of the size  of the sampled states used to estimate the dimensionality in Section 5.1? I am also curious about how different numbers affect the dimensionality estimated.*
>
> This is provided in Figure 5. We have provided how the estimate changes with the increasing number of samples and in the latest version we are also providing the estimate with sample sizes (with ‘o’ markers). We observe that, except for Walker2D environment, this estimate plateaus very quickly with 1000 samples. This is a pattern seen in the paper by Facco et al as well [3], that for some problems the estimate plateaus with fewer samples. Intuitively and empirically, this plateau with fewer samples phenomena happens when the sample is close to uniform from the manifold. We estimate the dimensionality with 10 different subsamples of the same size to provide an error region for the estimates (as described in Section 5.1) and as can be seen the estimate is consistent across different samples. Please let us know if this answers your question.

---

> > ### Author Response · Authors · 2022-11-14
> > **Response to Reviewer HpSc Continued**
> >
> > >*Similarly, personally, I am not satisfied with the results in Section 5.2, because I am very familiar with the performance of different representative DRL algorithms in MuJoCo and I do not think DDPG is a good baseline algorithm in MuJoCo. I recommend the authors to conduct the experiments based on TD3 or SAC which will be much more convincing, at least to me.*
> >
> > Thank you for pointing this out. We have now added the results for SAC and manifold representation learning in Appendix J (Figure 15). We have also provided architecture and hyperparameter details. In summary, we observe that our agent performs at par with the baseline in case of Walker2D, Cheetah and Reacher and slightly worse in the Swimmer environment. This was without an extensive hyperparameter tuning by varying the value of $\alpha_{\psi}$ over a very small range. Please let us know if this answers your query.
> >
> > >*For the comparison in Figure 5, do the authors have the results of a baseline that keeps the  $d_a + 1$ bottleneck structure while not uses $L_{\psi}$ (i.e., train as the original DDPG). I think this baseline will help a lot in understanding the efficacy of   $L_{\psi}$.*
> >
> > While we agree that a comparison with the case of keeping the architecture same as ours without the loss $L_{\psi}$ is a valid comparison, we do not agree with the idea that this should be the baseline to compare against. The reason being, a DNN might learn a mapping to a low-dimensional representation because of this bottleneck without explicit loss as shown by Laio et. al. [4]. While we are still running this experiment for 6 seeds, we can say from preliminary results that the agent performs worse than with the loss $L_{\psi}$. Our results will take a few more days since we have access to a limited number of GPUs, please watch out for a follow up response.
> >
> > >*For some additional discussion, I wonder the authors’ opinion on how this work, i.e., the manifold of effective state space, relates or differs to recent works on studying low-rank state (representation) space in DRL, e.g., [1,2]. *
> >
> > Thank you for pointing us to this very relevant and interesting work. Both of these works [1,2] utilize the underlying structure of Q-functions and the environment to provide better learning algorithms. One possible connection is the interplay of the low-dimensional structure of the state space with the low-rank feature of the $Q$-function matrix. Although most of the theoretical results in [2] are for a fixed dataset, we could imagine that adding the manifold hypothesis (the data is sampled from a low-dimensional manifold) to these results would reveal further theoretical details on how “rank collapse” can be prevented. We have now added this as related work.
> >
> > On minor issues:
> >
> > We have fixed the reference in Section 5.2
> > We have also fixed the font size and the resolution in the graphs
> >
> >
> > **References:**
> >
> >
> > [1] Yuzhe Yang, Guo Zhang, Zhi Xu, Dina Katabi. Harnessing Structures for Value-Based Planning and Reinforcement Learning. ICLR 2020
> >
> > [2] Aviral Kumar, Rishabh Agarwal, Dibya Ghosh, Sergey Levine. Implicit Under-Parameterization Inhibits Data-Efficient Deep Reinforcement Learning. ICLR 2021
> >
> > [3] Elena Facco, Maria d’Errico, Alex Rodriguez, and Alessandro Laio. Estimating the intrinsic dimension of datasets by a minimal neighborhood information. Scientific Reports, 7, 2017
> >
> > [4] Intrinsic dimension of data representations in deep neural networks, Alessio Ansuini, Alessandro Laio, Jakob H. Macke, Davide Zoccolan, Neurips 2019

---

> > > ### Author Response · Authors · 2022-11-15
> > > **Experiments with no manifold loss**
> > >
> > > Thank you for your continued patience. We have now added experiments for the scenario where there is no manifold loss but the architecture is still the same (with a bottleneck hidden layer of size $d_a + 1$), for the DDPG algorithm. This is presented in Appendix M. We observe that except for the Reacher environment all the environments perform worse without the bottleneck. In Reacher environment both the results are comparable. This furthers our argument that policy learning with an isometric mapping to low-dimensional manifolds is effective.
> > >
> > > We have also added a background sub-section for SAC (Appendix L.1) and also reported all the hyper-parameters used by us. Please take a look and let us know if this addresses all of your concerns.
> > >
> > > Once again our apologies for the delay and the staggered nature of our response, this was due to limited GPU compute availability.

---

> ### Author Response · Authors · 2022-11-23
> **Any further concerns or comments?**
>
> We have tried our best to address all the concerns raised by you. We have also added multiple new experiments, thanks to your suggestions. Please let us know if anything remains unanswered or unclear.

---

> > ### Author Response · Authors · 2022-12-05
> > **A Gentle Reminder for Further Comments**
> >
> > We deeply appreciate your comments and how they have gone a long way in improving our work. We would be grateful if you could make a pass through our new experiments (primarily Appendices L and M) and let us know if that addresses your concerns.

---

> ### Author Response · Authors · 2022-12-13
> **Another reminder for additional comments**
>
> Thank you very much for your valuable comments since they have significantly improved our work. We would be grateful if you could make a pass through the new experiments (primarily Appendices L and M) as requested by you and let us know if the latest revision addresses all your concerns.

---

### Author Response · Authors · 2022-11-16
**Summary of changes**

We summarize all the changes in the latest revision as compared to the original submission based on the comments and issues pointed out by the reviewers. We thank all the reviewers once again for their detailed comments and suggestions for improving our work.

Empirically validating assumptions and clarifying the applicability of our theoretical results:
1. We have empirically verified two assumptions: **i)** We empirically verify the deterministic nature of MujoCo environments, for all practical purposes, which is well known, **ii)** We verify the full rank assumption, to the best of our ability, over the transition function.
2. We have clarified how the other two assumptions, “no small orbits” and “action space restriction”, are reasonable, especially in physical environments.
3. We have asserted how our assumptions might seem restrictive at a first glance but are realistic, which is evident of the empirical validation of our results.

We have added further empirical evidence that an agent can learn effectively using this low-dimensional representation:
1. We apply the soft actor critic (SAC) baseline and compare it to an agent that learns a policy using SAC algorithm in tandem with manifold learning.
2. We have verified the efficacy of our manifold loss by comparing it to an architecture which has a bottleneck hidden layer of width $d_a +1$, for both DDPG and SAC.

These empirical results further our arguments. Not only is our theoretical result of interest to theorists and validates the manifold assumption made in the past, an agent can also learn effectively using this low-dimensional representation. This connects theory to practice for RL in continuous state and action spaces.

We have also made other minor changes:
1. Added to the related works section especially making a connection between recent theoretical works which assume a low-dimensional structure to data and our primary theoretical result.
2. We have fixed a minor error in the proof of Proposition B.2, the function $h$ from Proposition B.1 is equated to $\psi$ and not $\psi^{-1}$.
3. We have clarified the procedure for estimating the manifold loss in Section 5.2.
4. We have added a background section on SAC in the Appendix and also present the modified SAC algorithm which we employ.

---

### Decision · Program_Chairs · 2023-01-20

**Decision:**

Reject

**Justification For Why Not Higher Score:**

Paper is limited to deterministic MDPs, and there is a lack of experiments on truly high-dimensional MDPs like image-based observations.

**Justification For Why Not Lower Score:**

N/A

**Metareview: Summary, Strengths And Weaknesses:**

The paper presents a theoretically motivated analysis to understand the intrinsic/ underlying dimension of MDPs. The reviewers acknowledge the value in the proof along with the experimental analysis. However, there were significant concerns on the lack of discussion, the limited dimensionality of problems (state-based vs image-based) and limited applicability to deterministic systems. From an experimental standpoint, 6 seeds are too few hence does not substantiate the statistical claims in the paper. The variation in shaded regions in plots are large, which indicate the method completely fails in some runs.

As this paper was quite borderline, we had several discussions with the reviewers, ACs and SACs. Ultimately, the concerns empirical concerns raised by the reviewers were quite substantial and hence we lean on the side of rejection. Nevertheless, I encourage the authors to integrate the additional discussion and experimental suggestions made by the reviewers for a future submission.